# The 2023 National Offshore Wind data set (NOW-23)

Nicola Bodini[1], Mike Optis[2], Stephanie Redfern[1], David Rosencrans[3], Alex Rybchuk[1], Julie K. Lundquist[1,3,4], Vincent Pronk[1], Simon Castagneri[1], Avi Purkayastha[1], Caroline Draxl[1,4], Raghavendra Krishnamurthy[5], Ethan Young[1], Billy Roberts[1], Evan Rosenlieb[1], and Walter Musial[1]

[1]National Renewable Energy Laboratory, Golden, 80401, USA
[2]Veer Renewables, Courtenay, V9N 9B4, Canada
[3]Department of Atmospheric and Oceanic Sciences, University of Colorado, Boulder, 80303, USA
[4]Renewable and Sustainable Energy Institute, Boulder, 80303, USA
[5]Pacific Northwest National Laboratory, Richmond, 99354, USA

**Correspondence:** Nicola Bodini (nicola.bodini@nrel.gov)

**Abstract.** This article introduces the 2023 National Offshore Wind data set (NOW-23), which offers the latest wind resource information for offshore regions in the United States. NOW-23 supersedes, for its offshore component, the Wind Integration National Dataset (WIND) Toolkit, which was published a decade ago and is currently a primary resource for wind resource assessments and grid integration studies in the contiguous United States. By incorporating advancements in numerical weather prediction modeling, NOW-23 delivers an updated and cutting-edge product to stakeholders. In this article, we present the new data set, which underwent regional tuning and performance validation against available observations. We also provide a summary of the uncertainty quantification in NOW-23, along with NOW-WAKES, a 1-year post-construction data set that quantifies expected offshore wake effects in the U.S. mid-Atlantic lease areas. Stakeholders can access the NOW-23 data set at https://doi.org/10.25984/1821404 (Bodini et al., 2020).

*Copyright statement.* This work was authored in part by the National Renewable Energy Laboratory, operated by Alliance for Sustainable Energy, LLC, for the U.S. Department of Energy (DOE) under contract no. DE-AC36-08GO28308. Funding was provided by the U.S. Department of Energy Office of Energy Efficiency and Renewable Energy Wind Energy Technologies Office. Support for the work was also provided by the National Offshore Wind Research and Development Consortium under agreement no. CRD-19-16351 and by the Bureau of Ocean Energy Management under agreement number IAG-19-2123-4. The views expressed in the article do not necessarily represent the views of the DOE or the U.S. Government. The U.S. Government retains and the publisher, by accepting the article for publication, acknowledges that the U.S. Government retains a nonexclusive, paid-up, irrevocable, worldwide license to publish or reproduce the published form of this work, or allow others to do so, for U.S. Government purposes. Pacific Northwest National Laboratory (PNNL) is operated by Battelle Memorial Institute for the U.S. Department of Energy under Contract DE-AC05-76RL01830.

## 1 Introduction

In this article, we present the work done to create a state-of-the-art wind resource data set for all United States offshore regions (except for Alaska), called the 2023 National Offshore Wind data set (NOW-23). This work has been performed by





the National Renewable Energy Laboratory (NREL) and its partners, the University of Colorado Boulder and Veer Renewables.

In 2015, NREL produced the Wind Integration National Dataset (WIND) Toolkit (Draxl et al., 2015), a 7-year wind re-
source data set (2007–2013) covering the contiguous United States. The WIND Toolkit was built using the Weather Research
and Forecasting (WRF) mesoscale numerical weather prediction (NWP) model (Skamarock et al., 2019), and provided mod-
eled variables up to 200 m above the surface. Since its creation, the WIND Toolkit has become one of the most comprehensive
and commonly used data sets for wind resource assessment and grid integration studies in the United States, owing to the fact
it has been publicly available at no cost through Amazon Web Services. A wide variety of stakeholders, ranging from wind
energy developers and consultants, utilities, government organizations, and academic and research institutions, have taken ad-
vantage of the WIND Toolkit to foster wind energy development across the United States.

Since the release of the WIND Toolkit, extensive research in the field of NWP models has been completed, and many
advancements have been proposed and tested by the global atmospheric science community. Several field campaigns (e.g.,
Wilczak et al. (2015); Shaw et al. (2019); Fernando et al. (2019)) have been completed to collect observations useful to vali-
date and improve WRF capabilities. Growing research has assessed the sensitivity in the modeled wind resource to different
model inputs and parameterizations (e.g., Hahmann et al. (2020)). Also, new state-of-the-art reanalysis products (namely, ERA-
5 (Hersbach et al., 2020)) have been released and can now be used as boundary conditions to feed WRF. Finally, a broader
scientific consensus agrees that data sets of at least 20 years are needed for a robust quantification of the long-term wind re-
source at the site of interest.

Given the extensive success of the WIND Toolkit, NREL and its partners are committed to ensuring that the latest advance-
ments in the atmospheric modeling community are provided to stakeholders. As such, a next-generation product to replace
the WIND Toolkit is needed to ensure the most accurate wind resource data are given to the U.S. wind energy community.
Given the current and expected future sparsity of offshore hub-height observations, a national state-of-the-art mesoscale mod-
eled wind resource data set represents an even more critical need to support the breadth of offshore wind energy analyses and
stakeholders that rely on such data.

Here, we present NOW-23, a validated national offshore wind resource data set for the United States. The main final product
of this research effort is a WRF-based atlas of the offshore wind resource for all U.S. offshore waters, covering at least 20
years, which is made available at no cost to the public, with data at 5-minute temporal resolution, 2-km horizontal resolution,
and up to 500 m above the surface. In Section 2, we present the general modeling approach used to develop the NOW-
23 data set. Sections 3 through 10 describe the NOW-23 data set in each modeled offshore region. Section 11 describes
our uncertainty quantification efforts. Section 12 introduces NOW-WAKES, a post-construction data set for the mid-Atlantic
domain. In Section 13, we provide instructions on how to access the NOW-23 data set, followed by our main conclusions. In the
appendix, we provide additional analyses on the seasonal and diurnal variability of the modeled wind resource, the variability



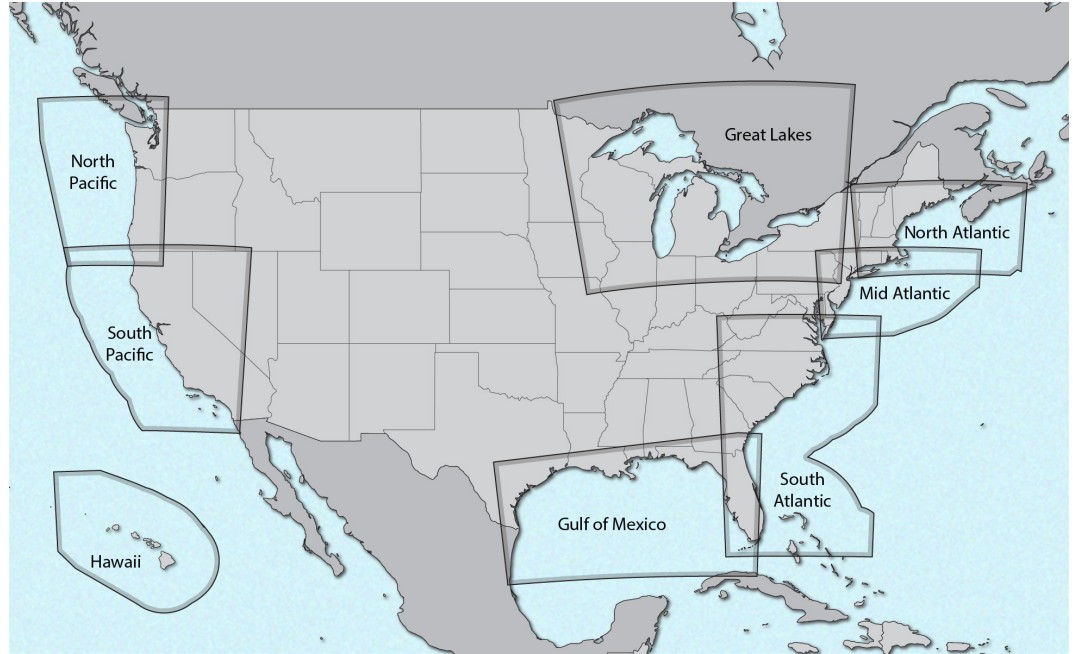

**Figure 1.** Map of the regional WRF domains used to build the NOW-23 data sets, masked at the limit of the exclusive economic zone of the countries with jurisdiction on the offshore regions being considered.

of the mean wind speed at overlapping boundaries of neighboring regional domains, and a comparison of the mean wind speed predictions with what was modeled by the previous-generation WIND Toolkit.

## 2 Description of the WRF simulations and general validation approach

To create the NOW-23 data set, we adopt a regional approach. For each offshore region, we perform a separate numerical simulation, whose setup is selected (in most regions) through validation against available observations, so that the model can be customized to account for regionally unique wind resource phenomena. Figure 1 shows the eight regional domains of the NOW-23 data set. We note that WRF domains have a rectangular shape. However, to limit the storage requirements for the NOW-23 data files, most of the regional data sets are masked (after the WRF simulations are done) based on the extension of

the exclusive economic zone (EEZ), which is the area where a country has jurisdiction over natural resources, including wind, roughly at 212 nautical miles from the coast.

Table 1 summarizes the common attributes for all the WRF simulations used to build NOW-23. We run the simulations in 1-month segments, and then concatenate them at each grid cell in the post-processing phase. In doing so, we consider a spin-up period of 2 days (for example, the May 2015 run actually starts on April 29, 2015) to let the model stabilize from the initial

conditions imposed to WRF. The choice of using 1-month segments is dictated by the need to have a limited number of restart





**Table 1.** WRF common attributes in all the simulations used to build the NOW-23 data set.

| Feature | Specification |
|---|---|
| **WRF version** | 4.2.1 |
| **Nesting** | 6 km, 2 km |
| **Vertical levels** | 61 |
| **Near-surface-level heights** | 12 m, 34 m, 52 m, 69 m, 86 m, 107 m, 134 m, 165 m, 200 m |
| **Atmospheric nudging** | Spectral nudging on 6-km domain, applied every 6 hours |
| **Microphysics** | Ferrier |
| **Longwave radiation** | Rapid Radiative Transfer Model |
| **Shortwave radiation** | Rapid Radiative Transfer Model |
| **Topographic database** | Global Multi-Resolution Terrain Elevation Data from the United States Geological Service and National Geospatial-Intelligence Agency |
| **Land-use data** | Moderate Resolution Imaging Spectroradiometer 30s |
| **Cumulus parameterization** | Kain-Fritsch |

periods for grid applications, as every restart might create false ramps, and by the desire to reduce the overall time needed to run the simulations, given the parallel computing capabilities offered by NREL's supercomputer where the simulations are performed. As will be detailed in later sections, we do not observe degraded performance with time in each calendar month.


Table 1 does not list some important attributes that need to be set before running WRF. In fact, many different setup choices need to be made before running a WRF simulation so that multiple simulations run with different setups will lead to a range of modeled conditions. It is therefore essential to tune the WRF setup to obtain accurate model predictions over the region of interest. For the majority of the regions considered in this work, we consider and run multiple WRF setups (i.e., a WRF ensemble) over a 1-year period and validate them against available observations to select the configuration that is best suited for long-term offshore wind resource assessment in each region. To determine which set of WRF setups to consider in our validation experiments, we leveraged recent research in the area to understand which choices strongly impact the WRF-modeled wind resource. A detailed list of the studies on the WRF-predicted wind speed sensitivity to the WRF setup is included in Optis et al. (2020c). Here, we highlight the exhaustive effort by Hahmann et al. (2020) to develop the New European Wind Atlas (NEWA) as well as an offshore analysis of the WRF model sensitivity that NREL recently completed in partnership with Rutgers University Center for Ocean Observing Leadership (Optis et al., 2020b). Based on the findings from recent literature on the topic, we identify the following five model setup choices as the most influential on the WRF-predicted wind resource: the reanalysis forcing product, the planetary boundary layer (PBL) scheme, the sea surface temperature (SST) product, the land surface model (LSM), and the surface layer scheme. For each region where we run a short-term WRF ensemble, we consider a subset of setups resulting from the combination of the following choices:





– Reanalysis forcing product: Reanalysis products are used as boundary conditions for WRF. We consider the state-of-the-art ERA5 reanalysis product developed by the European Centre for Medium-Range Weather Forecasts (Hersbach et al., 2020) and the Modern-Era Retrospective analysis for Research and Applications, Version 2 (MERRA-2, Gelaro et al. (2017)), developed by the National Aeronautics and Space Administration (NASA). Both these reanalysis products have been widely used in wind-energy-related applications and represent some of the most advanced reanalysis products available to date. Data from ERA5 are provided at hourly resolution and $0.25° \times 0.25°$ horizontal resolution. MERRA-2 also provides data at hourly resolution but at a coarser horizontal resolution, $0.50° \times 0.625°$.

– Planetary boundary layer scheme: The choice of the PBL scheme has critical consequences on how WRF will model turbulent exchanges in the atmospheric boundary layer, and it is expected to have a significant impact on the wind speed predictions. Here, we consider the Mellor–Yamada–Nakanishi–Niino (MYNN, Nakanishi and Niino (2009)) and the Yonsei University (YSU, Hong et al. (2006)) PBL schemes. These parameterizations are widely considered the two most popular PBL schemes in WRF, especially when considering wind-related applications; YSU was used in the WIND Toolkit, and MYNN in the NEWA (Dörenkämper et al., 2020).

– Sea surface temperature product: Because the focus of this research effort is on offshore wind, the choice of the sea surface temperature product, which acts as a lower boundary condition for WRF, should also be assessed in detail. The first SST product we consider is the Operational Sea Surface Temperature and Sea Ice Analysis (OSTIA) data set produced by the UK Met Office (Donlon et al., 2012; Hirahara et al., 2016), which provides data at $1/20°$ horizontal resolution and is the standard product included in both ERA5 and MERRA-2. Next, we consider the National Center for Environmental Prediction (NCEP) Real-Time Global SST product (Thiébaux et al., 2003), at $1/12°$ horizontal resolution.

– Land surface model: The choice of the land surface model can have a significant impact on offshore waters near the coast, as the LSM regulates the exchange of energy and water fluxes between the land surface and the atmosphere. We consider the Noah LSM and the updated Noah-Multiparameterization (Noah-MP) LSM (Niu et al., 2011).

– Surface layer scheme: Finally, the surface layer scheme handles how fluxes of heat, momentum, and moisture move from the surface to the boundary layer above. We consider the MM5 (Grell et al., 1994; Jiménez et al., 2012) and MYNN (Olson et al., 2021) surface layer schemes. Both are built on the Monin–Obukhov similarity theory, but the MYNN parameterization has been designed to specifically interface with the MYNN PBL scheme.

In the majority of the offshore regions modeled in NOW-23, we leverage available offshore or coastal observations to determine the best performing WRF setup. All the observational data sets used in the development of NOW-23 are shown in the map in Fig. 2 and will be described in detail in the next sections for each offshore region. In general, we use data from all the offshore lidars we have access to. We also consider observations collected by near-surface buoys from the National Data Buoy Center (NDBC) and coastal radars, as resources allow.



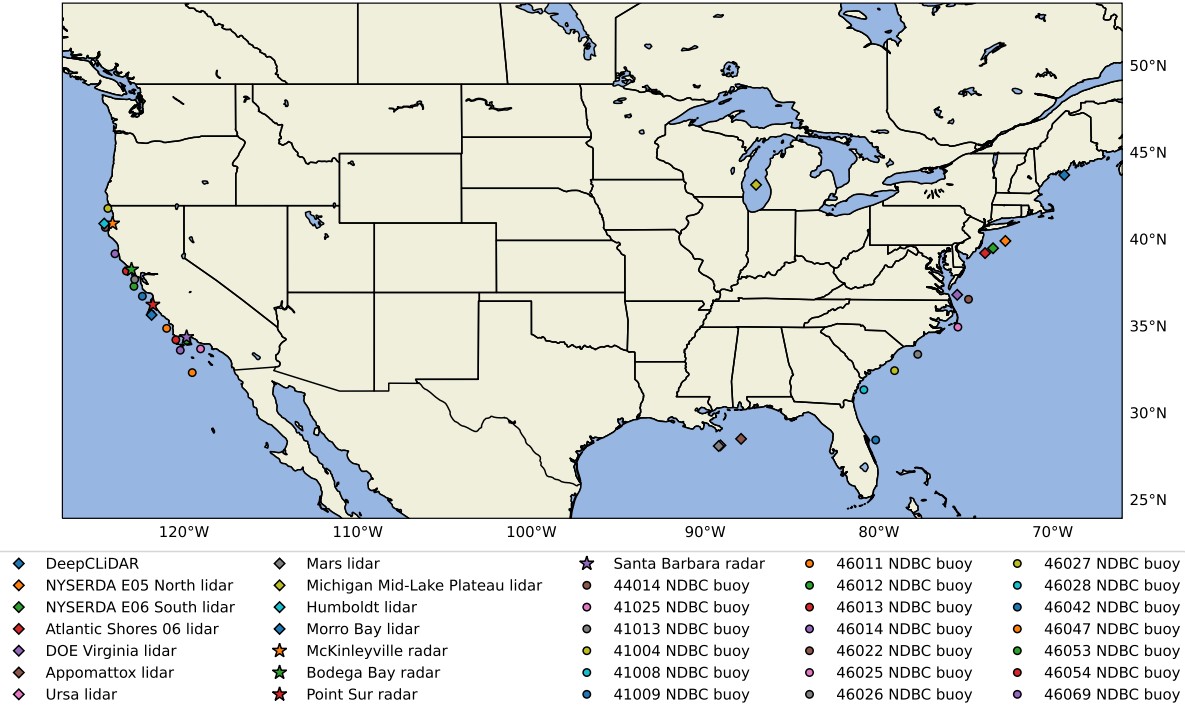

**Figure 2.** Map of the observational data sets used to validate the NOW-23 data set.

We base our validation approach on the best practices detailed in Optis et al. (2020a) and, whenever possible, use multiple error metrics between modeled and observed wind speed for our model validation and setup selection:

- Bias.

- Centered (or unbiased) root-mean-square error (cRMSE).

- Pearson's correlation coefficient (r).

- A comparison between the standard deviation of modeled and observed wind speed.

A perfect model setup would have zero bias, zero cRMSE, r=1, and a modeled wind speed standard deviation equal to that of the observed wind. To summarize the latter three metrics, we adopt the Taylor diagram (Taylor, 2001), which is a mathematical diagram that graphically summarizes model skills in terms of cRMSE, r, and standard deviation on a single plot.



**Table 2.** List of the 16 WRF ensemble members used for NOW-23 setup selection and validation in the mid-Atlantic region.

| WRF ensemble member name | Reanalysis product | PBL scheme | SST product | LSM | Surface layer scheme |
|---|---|---|---|---|---|
| WRF1 | ERA5 | MYNN | OSTIA | NOAH | MYNN |
| WRF2 | ERA5 | YSU | OSTIA | NOAH | MM5 |
| WRF3 | ERA5 | MYNN | OSTIA | NOAH-MP | MYNN |
| WRF4 | ERA5 | YSU | OSTIA | NOAH-MP | MM5 |
| WRF5 | ERA5 | MYNN | NCEP | NOAH | MYNN |
| WRF6 | ERA5 | YSU | NCEP | NOAH | MM5 |
| WRF7 | ERA5 | MYNN | NCEP | NOAH-MP | MYNN |
| WRF8 | ERA5 | YSU | NCEP | NOAH-MP | MM5 |
| WRF9 | MERRA-2 | MYNN | OSTIA | NOAH | MYNN |
| WRF10 | MERRA-2 | YSU | OSTIA | NOAH | MM5 |
| WRF11 | MERRA-2 | MYNN | OSTIA | NOAH-MP | MYNN |
| WRF12 | MERRA-2 | YSU | OSTIA | NOAH-MP | MM5 |
| WRF13 | MERRA-2 | MYNN | NCEP | NOAH | MYNN |
| WRF14 | MERRA-2 | YSU | NCEP | NOAH | MM5 |
| WRF15 | MERRA-2 | MYNN | NCEP | NOAH-MP | MYNN |
| WRF16 | MERRA-2 | YSU | NCEP | NOAH-MP | MM5 |

## 3  NOW-23 data set in the mid-Atlantic region

We start the description of the NOW-23 data set with the mid-Atlantic domain, an area with multiple publicly available observational data sets of hub-height offshore wind and therefore an ideal region to detail the validation approach adopted to develop NOW-23.

### 3.1  WRF ensemble members considered for long-term setup selection

For the mid-Atlantic region, we consider all the combinations resulting from the choices of reanalysis product, SST product, PBL scheme, and LSM detailed in Section 2. We do not consider for this region the impact of the surface layer scheme due to computational limitations; we use the MYNN surface layer option when the MYNN PBL scheme is used and the MM5 option with the YSU PBL scheme. These combinations result in 16 different WRF setups, which are all run over 1 year (using the general approach described in Section 2) from September 1, 2019, to August 31, 2020. Table 2 summarizes the 16 ensemble members used for WRF setup selection and model validation in this region.





## 3.2 Observations used for model validation

To select the best performing WRF setup in the region, which will be used for the long-term NOW-23 data set, we compare
modeled wind speed from the 16 WRF setups against observations collected by the three ZephIR ZX300M floating lidars,
shown in the map in Fig. 2. Two of the three lidars were deployed by the New York State Energy Research and Development
Authority (NYSERDA). The lidar on buoy E05 North is located at 39.97° N, 72.72° W; the buoy E06 South lidar is located at
39.55° N, 73.43° W. Wind speed and wind direction data for both lidars are available every 20 m from 20 m to 200 m above
sea level and are publicly distributed at 10-minute resolution after proprietary quality checks have been applied to the data.
For both lidars, we use observations collected between September 4, 2019, and August 31, 2020. The third lidar was deployed
by Atlantic Shores Offshore Wind closer to the coastline at 39.27° N, 73.88° W. This lidar measures wind speed every 20 m
from 40 m to 200 m above the surface. For this instrument, we leverage observations from February 26, 2020 (the start of its
deployment), to August 31, 2020.

## 3.3 Validation results

First, we assess the variability of the mean wind profiles across the 16 1-year WRF ensembles (Fig. 3). We color code the mean
wind profiles in terms of the PBL scheme used by the various ensemble members because, as will be detailed later, in some
offshore regions we see a significant deviation in modeled wind speeds between the MYNN and YSU PBL schemes. In the
mid-Atlantic, the WRF ensemble members that adopt the YSU PBL scheme model generally lower wind speeds compared to
MYNN at all three lidar locations, with the largest deviations occurring below 100 m above ground level (a.g.l.), with differ-
ences generally lower than 0.5 m s$^{-1}$, on average. While the MYNN generally slightly overestimates wind speeds below 50 m
a.g.l., we find that all the considered setups underestimate wind shear, thus resulting in a slight negative bias higher aloft.

We now dive deeper into the validation by assessing more quantitative performance metrics. All three lidars in this region
provide good measurements at a wide range of heights of interest for wind energy development. To capture the WRF model
performance across all heights, we perform our model validation in terms of the rotor equivalent wind speed (REWS), which is
the wind speed corresponding to the kinetic energy flux through a turbine's swept rotor area, when accounting for the vertical
shear. The REWS is calculated as

$$\text{REWS} = \left( \sum_{i=1}^{n} \text{WS}_i^3 \frac{A_i}{A} \right)^{1/3} \tag{1}$$

where $\text{WS}_i$ is the wind speed at the height level $i$, $n$ is the number of available heights across the wind turbine rotor disk, $A$ is
the whole area of the turbine rotor disk, and $A_i$ is the area of the $i$-th segment, i.e., the area for which $\text{WS}_i$ is representative.
Here, we consider the 10 MW turbine from Beiter et al. (2020) as representative of a typical commercial offshore turbine, with
a rotor diameter of 196 m and a hub height of 128 m. We consider horizontal layers in the turbine rotor disk equally spaced
every 20 m from 30 m to 190 m a.g.l., each associated with observed and modeled wind speed every 20 m from 40 m to 180
m a.g.l. Finally, the top layer extends from 190 m to 226 m and is associated with the 200-m wind speed, given the lack of

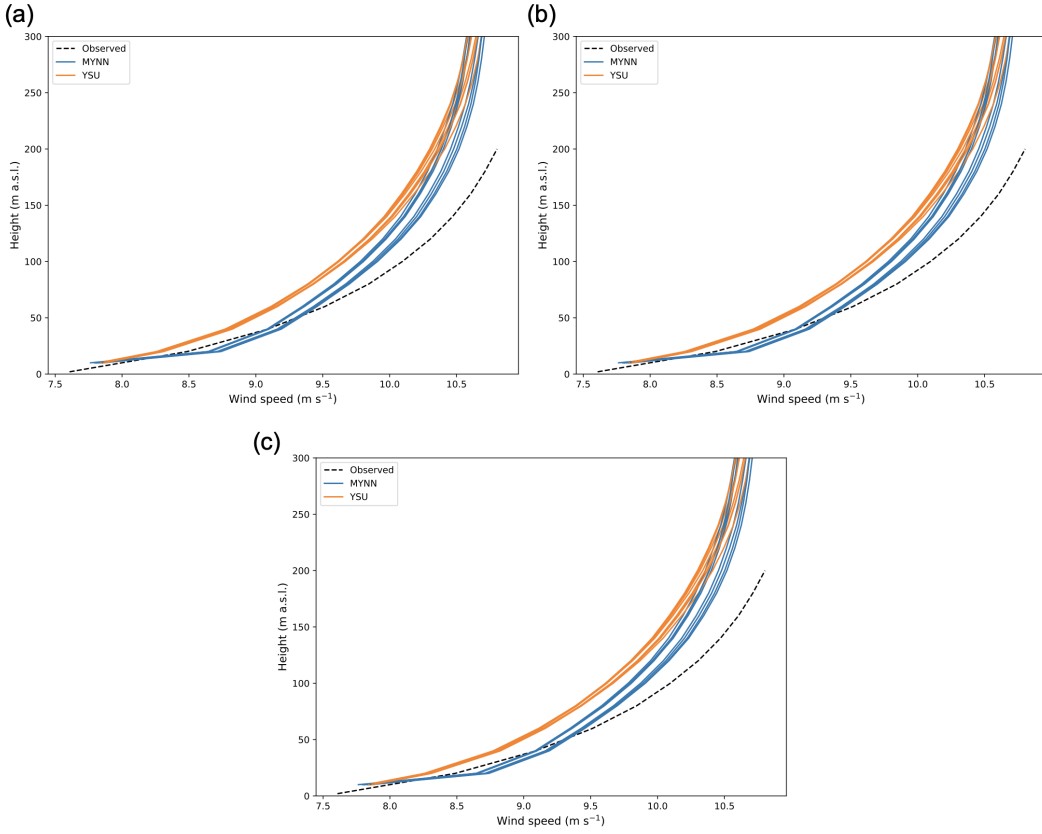

**Figure 3.** Mean wind speed profiles from the 16 WRF ensemble members and observed values at the location of the (a) NYSERDA E05 North lidar, (b) NYSERDA E06 South lidar, and (c) Atlantic Shores lidar.

lidar observations higher than 200 m a.g.l. While bigger turbines reaching higher heights are being installed in the region, the highest height of lidar observations at 200 m limits our ability to consider larger machines here.

We calculate bias, cRMSE, correlation, and standard deviation in terms of observed and modeled 10-minute REWS from all 16 WRF ensemble members and summarize results in the Taylor diagrams in Fig. 4. Ideally, a perfect member would be

represented in the Taylor diagram by a point on top of the black star in each diagram, which represents the observed values. At all three lidars, we find that the WRF ensemble members that use MERRA-2 as reanalysis forcing (WRF9 through WRF16, in red shades in the diagrams) show a significantly worse performance compared to the setups forced with ERA-5, with larger cRMSE and lower correlation. Also, the setups that adopt the YSU PBL scheme (even numbers) have a slightly better match with the standard deviation of the observed wind resource, but also a larger negative bias at all three offshore lidars compared

to the MYNN setups (odd numbers), as already noticed from the mean wind profile comparison above. The use of the NOAH

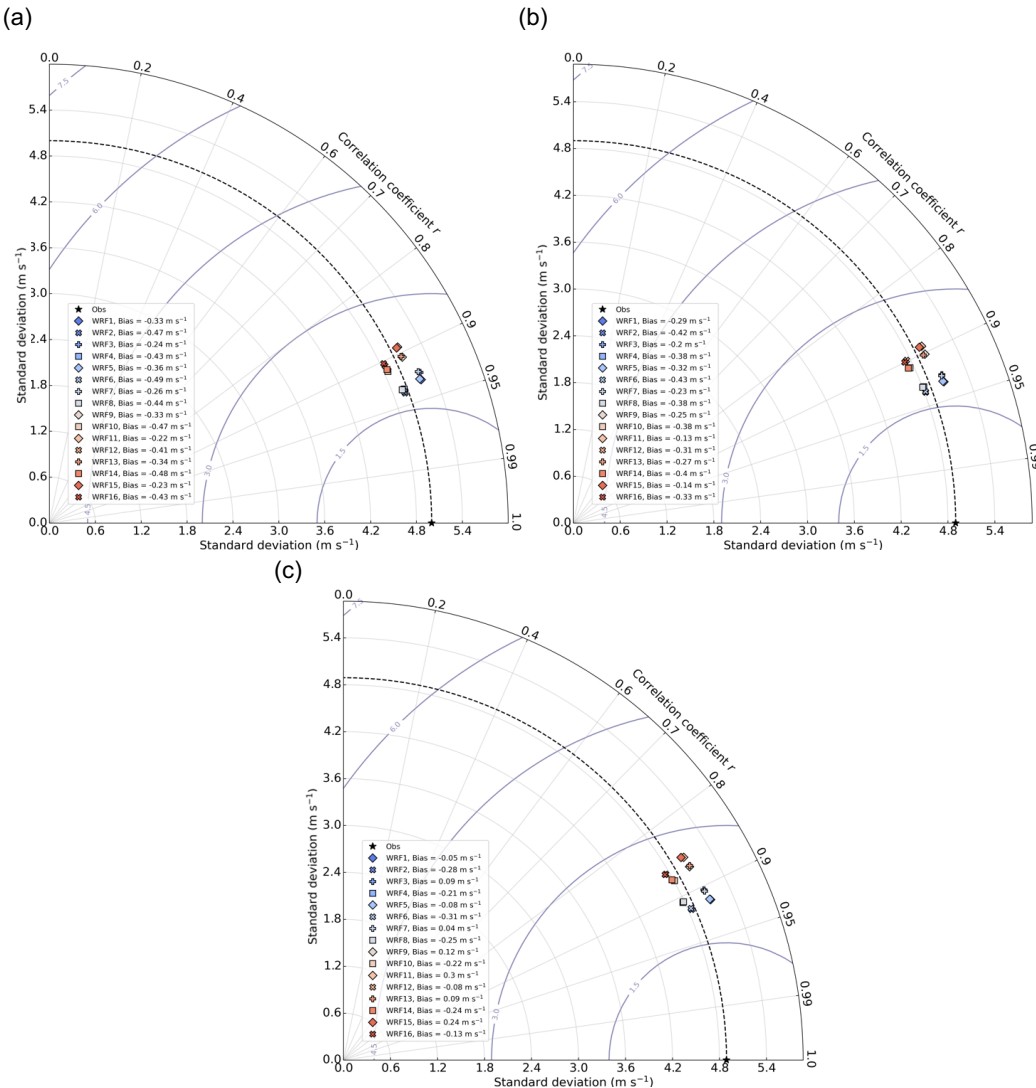

**Figure 4.** Taylor diagram of the 10-minute REWS from the 16 WRF ensemble setups at the location of the (a) NYSERDA E05 North lidar, (b) NYSERDA E06 South lidar, and (c) Atlantic Shores lidar.

LSM provides slightly better results than the NOAH-MP LSM in terms of correlation and cRMSE. Finally, we see that the choice of the SST product does not have a significant impact on the validation metrics.

Finally, we evaluate whether our choice of using a 1-month WRF re-initialization period has an impact on the model performance. To do so, we take the wind speed modeled by the 1-year WRF1 setup and check whether its performance against the observations from the two NYSERDA lidars gets worse in the latter part of each calendar month, again in terms of REWS.





**Table 3.** Selected WRF setup for the long-term NOW-23 data set in the mid-Atlantic region.

| WRF ensemble member name | Reanalysis product | PBL scheme | SST product | LSM | Surface layer scheme |
|:---:|:---:|:---:|:---:|:---:|:---:|
| WRF1 | ERA5 | MYNN | OSTIA | NOAH | MYNN |

For all metrics, we report no sign of performance degradation with time (figure not shown), which confirms the solidity of the modeling approach used.

### 3.4 Choice of the long-term WRF setup

The validation results across all three lidars show that WRF1 and WRF3 are the best performing setups in the region. While there is no clear winner between the two in terms of bias, WRF1 (i.e., the setup using the NOAH LSM) provides lower cRMSE and higher correlation compared to WRF3 (i.e., the setup using the NOAH-MP LSM), so we employ WRF1 for the long-term NOW-23 simulation (Table 3), which covers the period from January 1, 2000, to December 31, 2020. We note that additional details on the validation of the WRF1 model setup are provided in Pronk et al. (2022), where we assess the diurnal and annual variability of the WRF1 model setup performance, and compare it to the skills of the ERA-5 reanalysis product used to force WRF.

### 3.5 Long-term offshore wind resource

The 21-year mean wind speed at 160 m above sea level (a.s.l.) for the mid-Atlantic region is shown in Fig. 5. In Appendix A, we show the diurnal and seasonal variabilities of the long-term wind resource. The mean wind speed is stronger on the northeast portion of the domain, but the long-term averages are particularly good for offshore wind energy purposes in the whole extension of the domain.

## 4 NOW-23 data set in the North Atlantic region

We leverage the validation results from the mid-Atlantic region to infer conclusions about the WRF setup to use for the long-term wind resource modeling in the adjacent North Atlantic region, where we only have access to very limited hub-height observations of wind speed. Given this scenario, and considering the limited computational resources available, in this region we consider a single WRF setup, using the same choices selected for the creation of the NOW-23 data set in the mid-Atlantic region.

### 4.1 Observations used for model validation

No hub-height observational data sets are publicly available in this region. So, our model validation is constrained to values of mean 40 m and 100 m wind speeds taken from February 19, 2016, through October 28, 2016, at the DeepCLiDAR off the

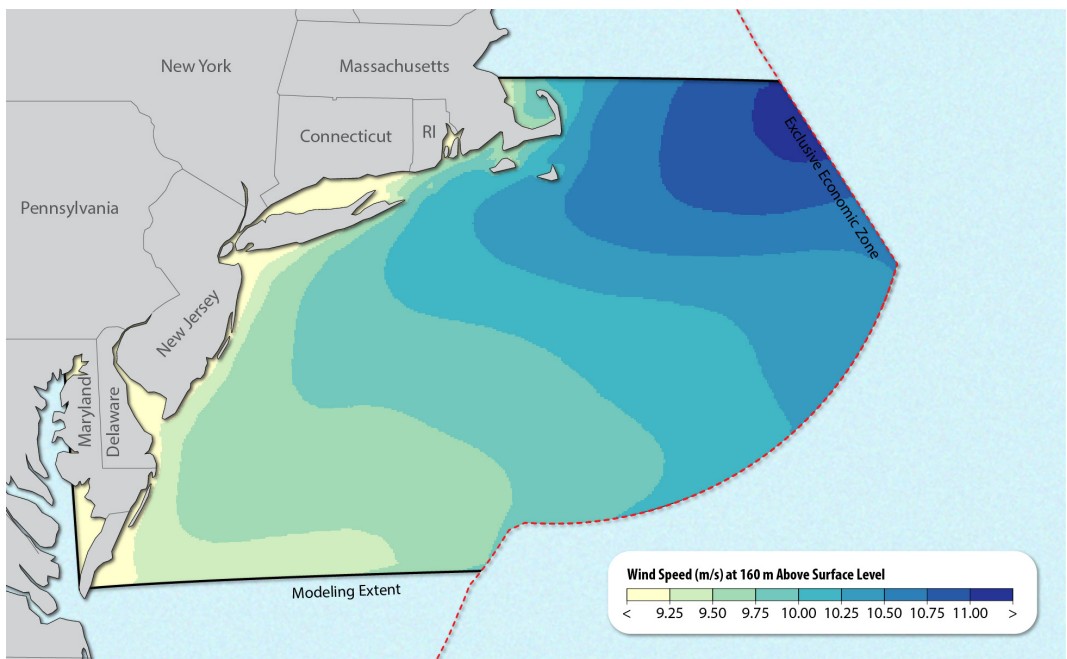

**Figure 5.** Map of the 21-year (2000–2020) mean wind speed at 160 m a.s.l. for the mid-Atlantic region. The red dashed line represents the limit of the U.S. EEZ. The continuous black line, where not overlaid with the EEZ boundary, shows the limit of the NOW-23 WRF domain.

215 Maine coast (43.77° N, 69.33° W, Fig. 2), 1.26 km west of Monhegan Island (Viselli et al., 2019, 2022). This floating lidar is a Windcube v2 offshore unit owned and operated by the University of Maine and samples wind speeds between 40 m and 200 m at 1 Hz resolution. The NOW-23 research team was not able to secure access to the raw lidar observations, so our validation is limited to the mean wind speed values as published in Viselli et al. (2022).

## 4.2 Validation results

220 Due to the particularly small sample size of available observations, we only compare the observed and modeled mean 40 m and 100 m wind speed at the DeepCLiDAR location over the deployment period of the lidar. We find good agreement between modeled and observed data at 100 m a.s.l., whereas the model slightly overestimates wind speed at 40 m a.s.l (Fig. 6).

## 4.3 Choice of the long-term WRF setup

The limited validation confirms that the chosen WRF setup has good agreement with the mean observations from the lidar at 225 the considered location, so we use the WRF1 setup to create the NOW-23 simulation in this region (Table 4), which covers the period from January 1, 2000, to December 31, 2020.





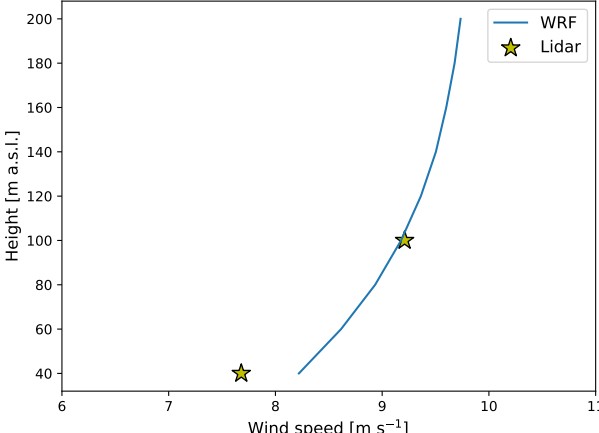

**Figure 6.** Mean wind speed profiles from the WRF run (WRF1 setup) and observed mean wind speed at 40 and 100 m a.s.l. at the location of the DeepCLiDAR buoy, from February 19, 2016, through October 28, 2016.

**Table 4.** Selected WRF setup for the long-term NOW-23 data set in the North Atlantic region.

| WRF ensemble member name | Reanalysis product | PBL scheme | SST product | LSM | Surface layer scheme |
|:---:|:---:|:---:|:---:|:---:|:---:|
| WRF1 | ERA5 | MYNN | OSTIA | NOAH | MYNN |

### 4.4 Long-term offshore wind resource

The 21-year mean wind speed at 160 m a.s.l. for the North Atlantic region is shown in Fig. 7. Its seasonal and diurnal vari-
abilities are shown in Appendix A. In this region, we find that the dominant gradient in mean wind speed is aligned with the
east-west direction, with stronger wind speeds observed further offshore. As observed in the mid-Atlantic region, the large
mean wind speed values make this region well suited for offshore wind energy.

## 5 NOW-23 data set in the South Atlantic region

### 5.1 WRF ensemble members considered for long-term setup selection

For the South Atlantic region, we consider 6 WRF setup combinations resulting from the choices of PBL scheme, surface layer
scheme, and LSM, as detailed in Section 2. We do not consider for this region the impact of the SST (we only consider the
OSTIA product, which has shown larger accuracy than the NCEP product based on the sensitivity analysis in the mid-Atlantic
region) and of the reanalysis product (because we found that ERA-5 has significantly better performance than MERRA-2 in
the mid-Atlantic region). We run all six setups across two sets of simulations (using the general approach described in Section

Earth System
Science
Data

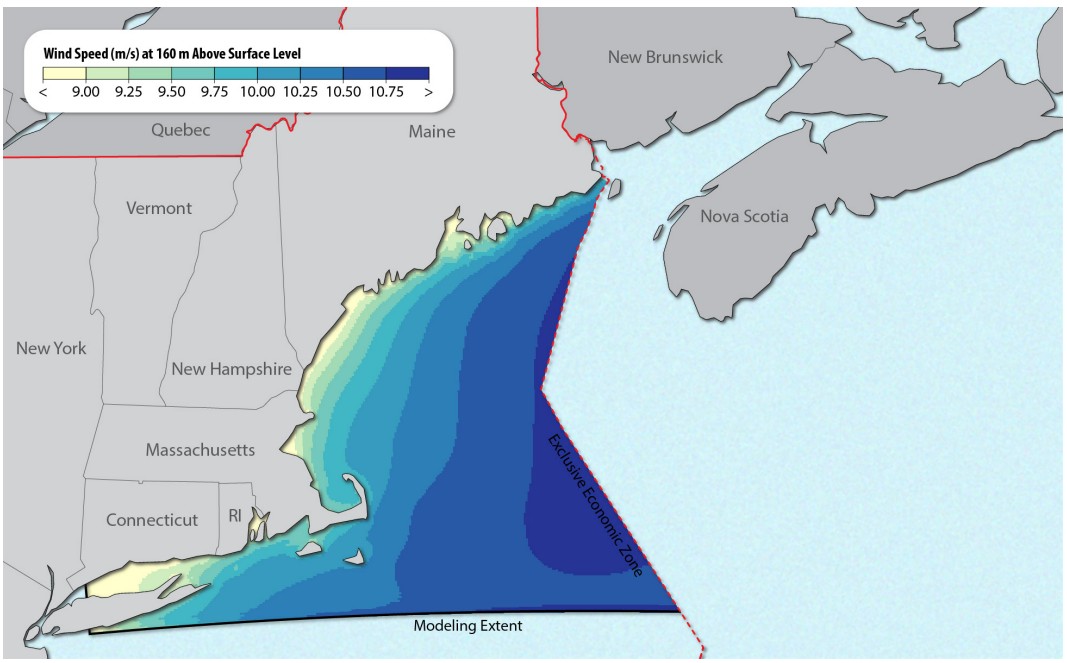

**Figure 7.** Map of the 21-year (2000–2020) mean wind speed at 160 m a.s.l. for the North Atlantic region. The red dashed line represents the limit of the U.S. EEZ. The continuous black line, where not overlaid with the EEZ boundary, shows the limit of the NOW-23 WRF domain.

**Table 5.** List of the six WRF ensemble members used for NOW-23 setup selection and validation in the South Atlantic region.

| WRF ensemble member name | Reanalysis product | PBL scheme | SST product | LSM | Surface layer scheme |
|---|---|---|---|---|---|
| WRF1 | ERA5 | MYNN | OSTIA | NOAH | MYNN |
| WRF2 | ERA5 | YSU | OSTIA | NOAH | MM5 |
| WRF3 | ERA5 | MYNN | OSTIA | NOAH-MP | MYNN |
| WRF4 | ERA5 | YSU | OSTIA | NOAH-MP | MM5 |
| WRF5 | ERA5 | MYNN | OSTIA | NOAH | MM5 |
| WRF6 | ERA5 | MYNN | OSTIA | NOAH-MP | MM5 |

2), one covering the whole year of 2015, and one covering the period from June 1, 2020, to December 31, 2020. Table 5 lists

the six ensemble members used for model setup selection and validation in this region.

### 5.2   Observations used for model validation

The main validation data set we use in this region is represented by observations collected by the U.S. Department of Energy (DOE) lidar located off the Virginia coast (36.87° N, 75.49° W) (Shaw et al., 2020). This lidar recorded observations from December 11, 2014, to May 31, 2016. For our validation, we use data from the whole year of 2015 to ensure all seasons are





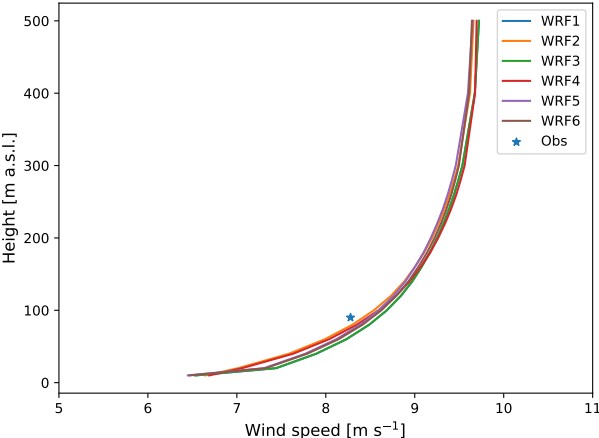

**Figure 8.** Mean modeled wind speed profiles over 2015 from the six WRF setups considered for the sensitivity analysis, and mean observed winds at 90 m from the DOE Virginia lidar.

equally represented. Conversations with the instrument mentors at Pacific Northwest National Laboratory (PNNL) revealed that only the lidar measurements at 90 m are unaffected by biases, so we limit the NOW-23 model validation to that height. Whereas algorithms to bias-correct the lidar measurements at other heights have been developed, we prefer not to use them here to avoid introducing additional uncertainty in the validation.

Additionally, the Avangrid company performed a validation using proprietary data from their Kitty Hawk North lidar, covering
the period from June 1 to December 31, 2020. Finally, we note how both the Virginia and Kitty Hawk North lidars are near the northern edge of our regional domain. Therefore, to improve the spatial coverage of the NOW-23 validation in the region, we leverage observations from six NDBC buoys across the domain to validate modeled near-surface atmospheric stability, quantified in terms of the difference between air temperature and sea surface temperature, over 2015.

### 5.3  Validation results

Figure 8 shows the mean modeled wind profiles over 2015 at the location of the Virginia lidar for the six WRF setups, as well as the mean 90 m wind speed from the DOE Virginia lidar over the same year. In general, a limited spread between the different WRF setups appears, and all setups provide a limited overestimation of mean wind speed at 90 m a.s.l. compared to the lidar observations.

We formalize the regional validation with the Taylor diagram in Fig. 9, again at 90 m a.s.l, using wind speed at 10-minute
resolution. The diagram reveals that the WRF2 setup performs best in the South Atlantic domain, with a low bias of +0.11 m s$^{-1}$, the highest Pearson's correlation coefficient (r = 0.83), lowest cRMSE (2.44 m s$^{-1}$), and the best match with observed wind speed standard deviation across all the WRF ensemble members.

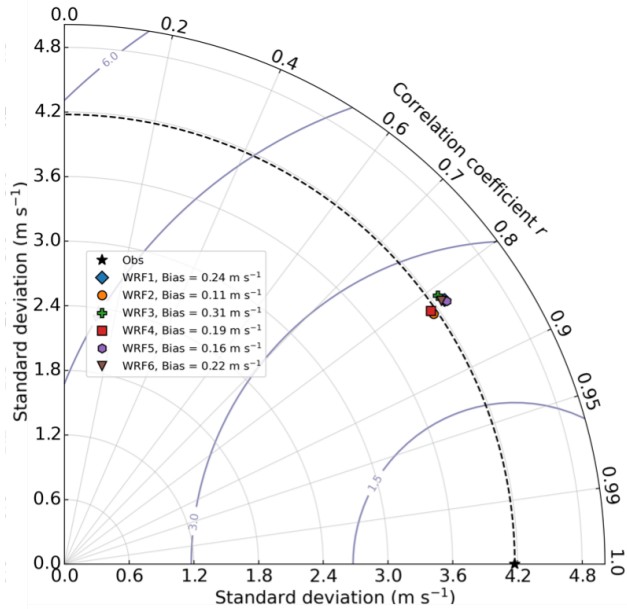

**Figure 9.** Taylor diagram of the 90 m wind speed from the six WRF ensemble setups at the location of the DOE Virginia lidar using 10-minute data.

Avangrid's Kitty Hawk North lidar data are proprietary, and therefore only the following qualitative results were shared with the NOW-23 team after the company compared the six WRF setups against their lidar observations over the last seven months of 2020:


- All WRF ensemble members performed well in terms of their wind rose.

- Overall, ensemble members WRF2 and WRF4 performed the best for both wind speed and wind shear profiles.

- Some specific months were challenging to model for all considered ensemble members.

Next, we leverage the more extensive spatial coverage of the NDBC buoys in the region to validate modeled atmospheric

stability near the surface, and we summarize these results in Table 6. We consider a positive difference between air temperature (at ∼4 m a.s.l. for the NDBC buoys, 2 m a.s.l. for the WRF simulations) and sea surface temperature as a proxy for stable conditions, and report its observed and modeled temporal frequency over 2015 in the table. The WRF setups that use MYNN as a PBL scheme (WRF1, WRF3, WRF5, and WRF6) overestimate atmospheric stability. On the other hand, YSU-modeled stability (WRF2 and WRF4) is generally more aligned with observations across the whole region. This result, if confirmed at

hub heights, is consistent with the larger wind speed bias the MYNN setups have in the Taylor diagram. In fact, under stable conditions, winds aloft can decouple from surface effects and greatly accelerate.





**Table 6.** Frequency of near-surface stable conditions from NDBC buoy observations and the six WRF ensemble members over 2015.

| NDBC buoy | Observed | WRF1 | WRF2 | WRF3 | WRF4 | WRF5 | WRF6 |
|---|---|---|---|---|---|---|---|
| 44014 | 34% | 34% | 33% | 32% | 33% | 35% | 35% |
| 41025 | 8% | 10% | 7% | 12% | 8% | 11% | 13% |
| 41013 | 20% | 23% | 20% | 24% | 20% | 22% | 25% |
| 41004 | 15% | 19% | 15% | 19% | 18% | 19% | 23% |
| 41008 | 34% | 37% | 32% | 35% | 35% | 37% | 41% |
| 41009 | 19% | 22% | 17% | 18% | 18% | 22% | 24% |

**Table 7.** Selected WRF setup for the long-term NOW-23 data set in the South Atlantic region.

| WRF ensemble member name | Reanalysis product | PBL scheme | SST product | LSM | Surface layer scheme |
|---|---|---|---|---|---|
| WRF2 | ERA5 | YSU | OSTIA | NOAH | MM5 |

Finally, as done in the mid-Atlantic domain, we check for an impact of using a 1-month WRF re-initialization period in our simulations. To do so, we take the WRF2 setup for 2015 and check whether its performance against the DOE Virginia lidar observations gets worse in the latter part of each calendar month. For all metrics, we see no sign of performance degradation with time (figure not shown), as already observed further north along the mid-Atlantic coast.

### 5.4 Choice of the long-term WRF setup

Our validation analysis across all considered instruments and atmospheric variables reveals that the WRF2 setup is the best preforming one in this region, and therefore we use this setup for the NOW-23 long-term simulation (Table 7), which covers the period from January 1, 2000, to December 31, 2020. We note that this setup differs from the setup used in the adjacent mid-Atlantic region, so that some discontinuity at the interface between the two domains is expected. Such a difference should, however, be limited in magnitude given the minimal difference between the mean wind profiles near the northern edge of the domain in Fig. 8. This discrepancy is described in further detail in Appendix B.

### 5.5 Long-term offshore wind resource

The 21-year mean wind speed at 160 m a.s.l. for the South Atlantic region is shown in Fig. 10. Wind speed is, on average, stronger in the northern half of the domain, with a clear north-south gradient that leads to mean differences of more than 2 m s$^{-1}$ between the northern and southern edges of the modeled region. We note that we had to limit the extent of the WRF domain for this large region to reduce computational requirements so that the northern portion of the domain does not reach the limit of the U.S. EEZ. Appendix A shows the diurnal and seasonal variability of the 160 m wind resource.



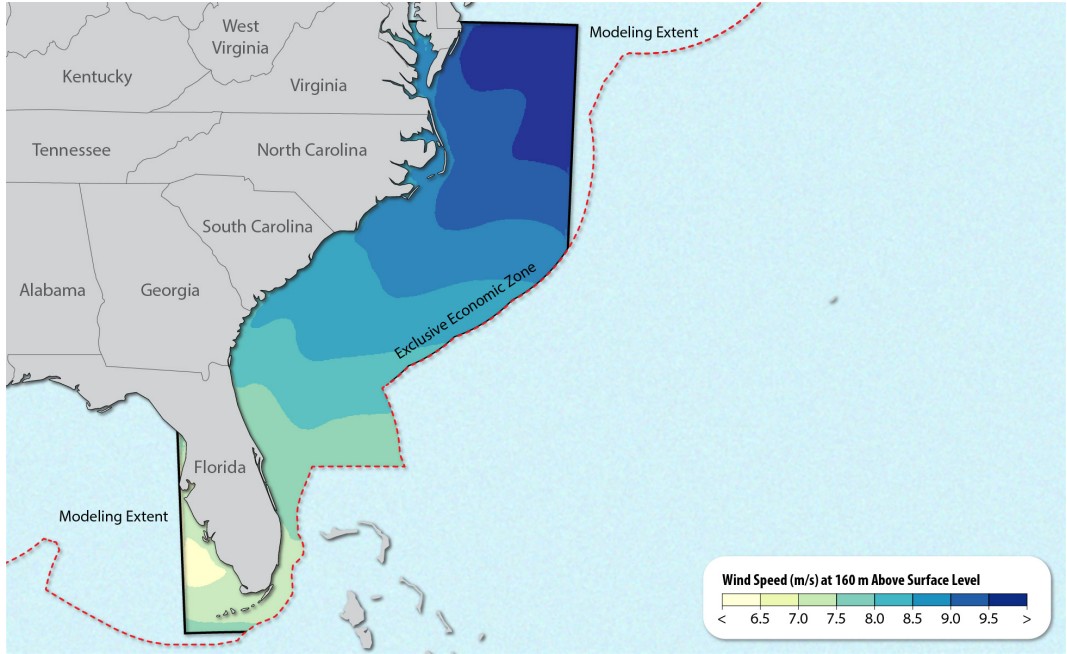

**Figure 10.** Map of the 21-year (2000–2020) mean wind speed at 160 m a.s.l. for the South Atlantic region. The red dashed line represents the limit of the U.S. EEZ. The continuous black line, where not overlaid with the EEZ boundary, shows the limit of the NOW-23 WRF domain.

**Table 8.** List of the six WRF ensemble members used for NOW-23 setup selection and validation in the Gulf of Mexico region.

| WRF ensemble member name | Reanalysis product | PBL scheme | SST product | LSM | Surface layer scheme |
|:---:|:---:|:---:|:---:|:---:|:---:|
| WRF1 | ERA5 | MYNN | OSTIA | NOAH | MYNN |
| WRF2 | ERA5 | YSU | OSTIA | NOAH | MM5 |
| WRF3 | ERA5 | MYNN | OSTIA | NOAH-MP | MYNN |
| WRF4 | ERA5 | YSU | OSTIA | NOAH-MP | MM5 |
| WRF5 | ERA5 | MYNN | OSTIA | NOAH | MM5 |
| WRF6 | ERA5 | MYNN | OSTIA | NOAH-MP | MM5 |

## 6  NOW-23 data set in the Gulf of Mexico

### 6.1  WRF ensemble members considered for long-term setup selection

For the Gulf of Mexico, we consider six ensemble members (Table 8). We only consider ERA-5 as reanalysis forcing and OSTIA as SST product for the same considerations listed for the South Atlantic domain. We test both the MYNN and YSU PBL schemes, each associated with the MYNN and MM5 surface layer scheme, respectively. We also consider the impact of the NOAH and NOAH-MP LSMs. We run the six ensemble members over year 2020.





**Table 9.** Selected WRF setup for the long-term NOW-23 data set in the Gulf of Mexico region.

| WRF ensemble member name | Reanalysis product | PBL scheme | SST product | LSM | Surface layer scheme |
|:---:|:---:|:---:|:---:|:---:|:---:|
| WRF2 | ERA5 | YSU | OSTIA | NOAH | MM5 |

### 6.2 Observations used for model validation

Observational wind speed data needed for model setup choice and validation were provided by Shell at three different ZX300 lidars south of the Louisiana coast (Fig. 2). These lidars, named Mars, Ursa, and Appomattox, are located on separate floating oil platforms ∼50 m above the ocean and sample winds up through ∼150 m above the ocean surface. Data are available from
305    Mars and Ursa for all of 2020, whereas measurements at Appomattox are only available from January through August 2020.

### 6.3 Validation results

Several buildings and structures are present on the Shell oil platforms. To minimize their wake impacts on the lidar observations, we perform our validation at 140 m a.s.l. (i.e., the maximum common height between the lidars and the WRF simulations). Figure 11 shows the Taylor diagrams at the three lidars using 10-minute data. At all three locations, the WRF2 setup outper-
310    forms the other five simulation setups, as it has the smallest bias (with still a limited tendency to under-forecast wind speeds), the smallest cRMSE, and the highest correlation (between 0.7 and 0.8 at each lidar).

### 6.4 Choice of the long-term WRF setup

Our validation analysis shows that the WRF2 setup is the best preforming configuration in this region, and therefore we use this setup for the long-term NOW-23 simulation (Table 9), which covers the period from January 1, 2000, to December 31,
315    2020.

### 6.5 Long-term offshore wind resource

The 21-year mean wind speed at 160 m a.s.l. for the Gulf of Mexico region is shown in Fig. 12, and Appendix A shows the diurnal and seasonal variability of the modeled wind resource. We observe a clear east-west gradient, with faster winds on the western side of the Gulf and slower winds on the eastern side. This general pattern has also been observed in other studies
320    (de Velasco and Winant, 1996; Zavala-Hidalgo et al., 2014).



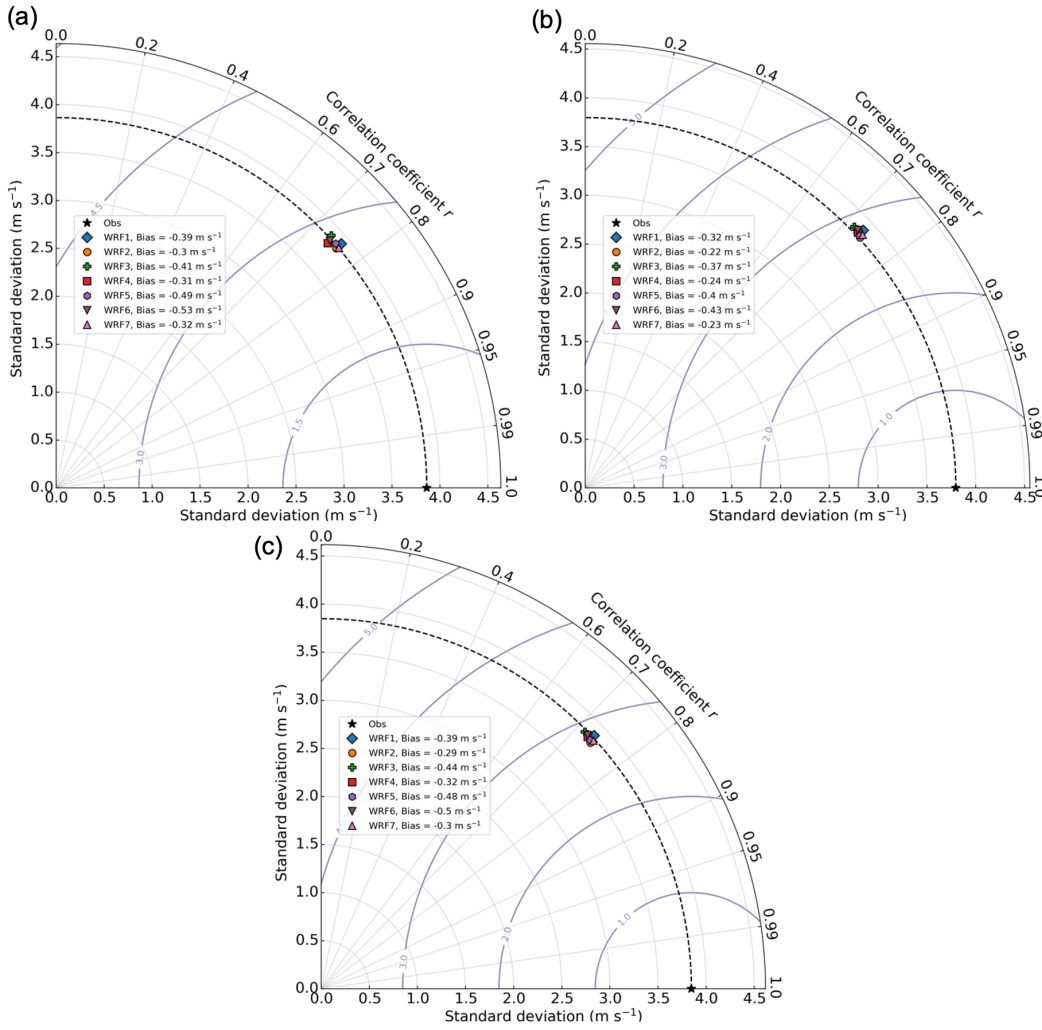

**Figure 11.** Taylor diagram of the 140 m wind speed from the six WRF ensemble setups at the location of the (a) Appomattox, (b) Ursa, and (c) Mars lidars, using 10-minute data.

## 7 NOW-23 data set in the Great Lakes

### 7.1 WRF ensemble members considered for long-term setup selection

For the Great Lakes, we only consider three ensemble members (Table 10) by leveraging the results of the model validation in the other offshore regions. Therefore, we only consider ERA-5 as reanalysis forcing and OSTIA as SST product. We test both the MYNN and YSU PBL schemes, each associated with the MYNN and MM5 surface layer scheme, respectively. We also consider the impact of the NOAH and NOAH-MP LSMs. We run the three ensemble members for year 2012.





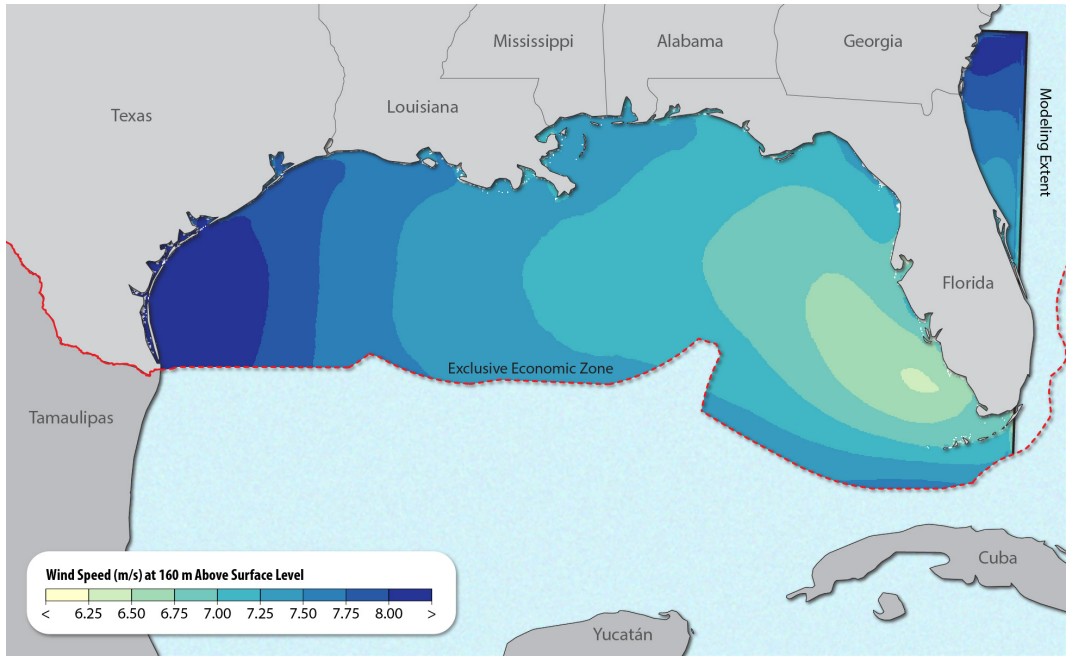

**Figure 12.** Map of the 21-year (2000–2020) mean wind speed at 160 m a.s.l. for the Gulf of Mexico region. The red dashed line represents the limit of the U.S. EEZ. The continuous black line, where not overlaid with the EEZ boundary, shows the limit of the NOW-23 WRF domain.

**Table 10.** List of the three WRF ensemble members used for NOW-23 setup selection and validation in the Great Lakes region.

| WRF ensemble member name | Reanalysis product | PBL scheme | SST product | LSM | Surface layer scheme |
|:---:|:---:|:---:|:---:|:---:|:---:|
| WRF1 | ERA5 | MYNN | OSTIA | NOAH | MYNN |
| WRF2 | ERA5 | YSU | OSTIA | NOAH | MM5 |
| WRF3 | ERA5 | YSU | OSTIA | NOAH-MP | MM5 |

### 7.2 Observations used for model validation

To select the best performing model setup in this region, we leverage observations from one lidar, whose location is shown in the map in Fig. 2. The lidar was deployed in the middle of Lake Michigan and measured wind speed at 75, 90, 105, 125, 150, and 175 m above the surface. For this instrument, we use 10-minute average observations from May 8, 2012, to December 17, 2012. We discard data at 175 m, as they are deemed unrealistic.



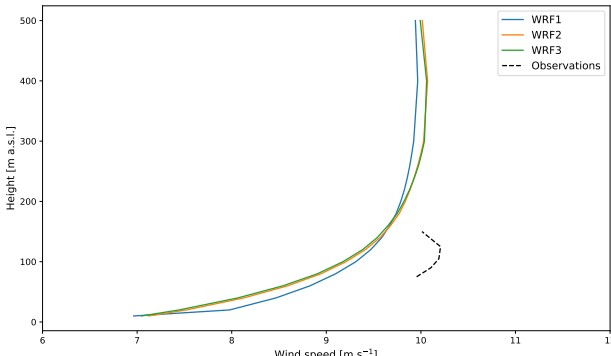

**Figure 13.** Mean wind speed profiles over the period from May 8, 2012, to December 17, 2012, from the three WRF ensemble members and lidar observations at the location of the Lake Michigan lidar.

**Table 11.** Selected WRF setup for the long-term NOW-23 data set in the Great Lakes region.

| WRF ensemble member name | Reanalysis product | PBL scheme | SST product | LSM | Surface layer scheme |
|---|---|---|---|---|---|
| WRF1 | ERA5 | MYNN | OSTIA | NOAH | MYNN |

## 7.3 Validation results

Figure 13 shows the mean wind profiles from the three WRF ensemble members and the Michigan Lake lidar during the period of record of the lidar observations. All three WRF setups underestimate wind speed compared to the lidar observations. The setup using the MYNN PBL scheme predicts higher wind speeds compared to the YSU setups in the lowest 200 m, whereas YSU models stronger winds higher aloft. In any case, the difference between all models is rather limited at all heights. We see slight differences between the WRF2 and WRF3 setups, an additional indication that using the NOAH or NOAH-MP LSMs has a limited impact on the mean wind speed profiles, as observed for the other offshore regions.

We next consider the Taylor diagram (Fig. 14) at 105 m, using 10-minute data. For the modeled data, we linearly interpolate wind speed at 100 m and 120 m to allow for a direct comparison with the lidar observations. All three considered WRF setups show similar performance in terms of bias, correlation and cRMSE, but the WRF1 setup shows significantly better results in terms of the comparison with the standard deviation of the observed wind speed.

## 7.4 Choice of the long-term WRF setup

Our validation analysis shows that the WRF1 setup is the best preforming configuration in this region, and therefore we use this setup for the NOW-23 long-term simulation (Table 11), which covers the period from January 1, 2000, to December 31, 2020.

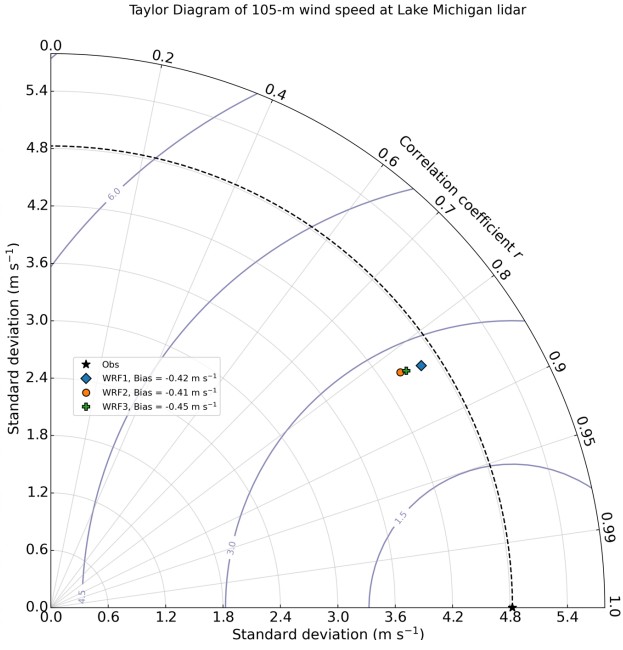

**Figure 14.** Taylor diagram of the 105 m wind speed from the three WRF ensemble setups at the location of the Lake Michigan lidar, at 10-minute resolution.

## 7.5 Long-term offshore wind resource

The 21-year mean wind speed at 160 m a.s.l. for the Great Lakes region is shown in Fig. 15. Wind speed gets stronger near the center of the lakes, especially for the larger Lakes Michigan, Superior, and Huron. The magnitude of the mean wind speed
across the domain is similar to what is found in the northern portion of the U.S. East Coast. Appendix A describes the diurnal and seasonal variability of the long-term modeled wind resource.

## 8   NOW-23 data set in the South Pacific region

### 8.1   Background on the South Pacific data set

The South Pacific (offshore California) was the first region to be considered for the development of this long-term wind re-
source data set as part of a BOEM-funded pilot project, and the development of the data set has been subject to revisions. The development of the first version of a 20-year data set for the California Outer Continental Shelf (OCS), called "CA20," is described in Optis et al. (2020d). As detailed in the report, 16 WRF ensemble members were considered by tweaking the reanalysis forcing, PBL scheme, SST product, and LSM. The WRF setup employed for the long-term CA20 data set was selected based on a validation against available observations at that time (an array of near-surface NDBC buoys and coastal
radars) and based on the validation results obtained in the mid-Atlantic region. As a result of this validation, a 20-year data set

Earth System
Open Access    Science    Discussions
Data

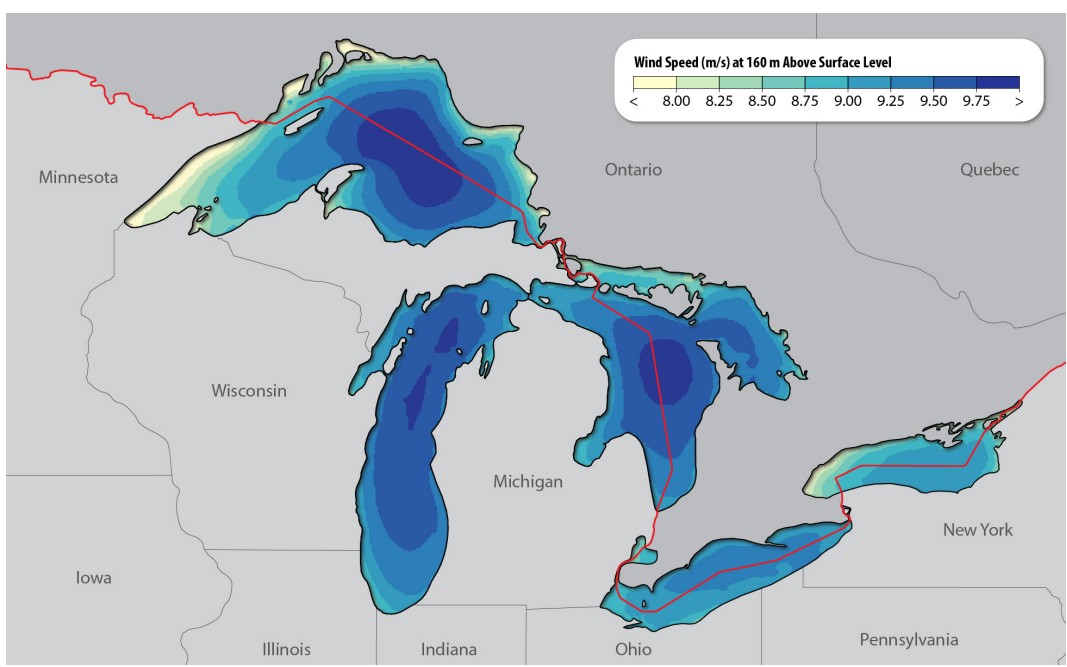

**Figure 15.** Map of the 21-year (2000–2020) mean wind speed at 160 m a.s.l. for the Great Lakes region. The red dashed line represents the limit of the U.S. EEZ.

(January 1, 2000–December 31, 2019) was run using the ERA-5 reanalysis product, the MYNN PBL scheme, the OSTIA SST product, the NOAH LSM, and the MYNN surface layer scheme.

The available measurements used to validate the CA20 model are less than ideal. In fact, the NDBC buoys only provide
measurements close to the surface, which are insufficient to determine the wind resource at the relevant heights for wind energy purposes. Coastal radars provide measurements at heights of interests, but the WRF validation at their locations becomes uncertain due to large meteorological gradients at the land-ocean interface. Also, results from the mid-Atlantic validation cannot be directly applied to the U.S. West Coast, given the different domain-specific processes and features that might determine a different optimal WRF setup. When the CA20 data set was developed, the absence of floating lidar observations in the Cali-
fornia OCS was recognized as a significant limitation to the analysis and initial validation of CA20.

Two floating lidars were deployed in the region in late 2020—one near the Humboldt wind energy lease area in the northern part of the domain, and one near the Morro Bay wind energy lease area further south (Krishnamurthy et al., 2023). The WRF setup originally used in the CA20 data set was then run over the October 2020–September 2021 period and compared against
the concurrent observations collected by the two lidars. This comparison revealed a significant bias in the CA20 modeled data, especially at the Humboldt lidar location. This bias and its impact on energy assessments in the California OCS are described



in Bodini et al. (2022).

Additional analysis has shown that the choice of the PBL scheme is responsible for the vast majority of the bias in the CA20
data set. The MYNN PBL scheme overestimates atmospheric stability, especially at Humboldt, resulting in reduced vertical
turbulent mixing, allowing for the acceleration of hub-height winds, more intense low-level jets, and higher-amplitude inertial
oscillations. Also, during synoptic-scale northerly flows driven by the North Pacific High and inland thermal low, simulations
using the MYNN PBL scheme show a coastal warm bias in selected case studies, which contributes to the modeled wind speed
bias by altering the boundary layer thermodynamics via a thermal wind mechanism. Further, we found that the YSU PBL
scheme strongly reduces the bias at both Humboldt and Morro Bay. Given the strong performance of the YSU-based runs in
the South Pacific region, we reran the full long-term simulation in the region using the YSU PBL scheme, and this version is
the one included in the NOW-23 data set. The results of this in-depth analysis, additional validation against floating lidars and
coastal radars, and description of the updated data set will be presented in an upcoming NREL report. Here, we present a short
summary of the main characteristics of the final NOW-23 data set for the South Pacific region for consistency with what is
described for the other offshore regions in the NOW-23 data set.

## 8.2 WRF ensemble members considered for long-term setup selection

As detailed in Optis et al. (2020d), the 16 WRF ensemble members listed in Table 12 were considered in the original develop-
ment of the CA20 data set. These 16 WRF simulations were run for year 2017.

After the Humboldt and Morro Bay floating lidars were deployed, the two ensemble members listed in Table 13 were run
from October 1, 2020, to September 30, 2021, to overlap with observations collected by the two floating lidars, as described in
Bodini et al. (2022) and an upcoming report on the South Pacific data set.

## 8.3 Observations used for model validation

Several observational data sets have been used to select the best performing model setup in this region, in both the creation of
the first-generation CA20 data set and the updated NOW-23 one. As described in Optis et al. (2020d), the first CA20 analysis
focused on four coastal wind profilers and an array of NDBC buoys in the region. On the other hand, the more recent analysis
that led to the identification and correction of the bias in the CA20 data set leveraged the two floating lidars at Humboldt and
Morro Bay (Krishnamurthy et al., 2023) as well as the coastal wind profilers at McKinleyville and Bodega Bay (Fig. 2). Details
about the instruments can be found in the reports that describe the validation efforts, as listed at the beginning of this section.

## 8.4 Validation results

We refer to the reports listed above for a comprehensive description of the results of the validation performed at the various
stages of the development of the South Pacific data set. Here, we only include in Fig. 16 the vertical profiles of bias, cRMSE,
and r for the WRF4 setup, which is chosen for the final NOW-23 data set, at the locations of the Humboldt and Morro Bay



**Table 12.** List of the 16 WRF ensemble members used for the initial long-term setup selection and model validation in the South Pacific region (for the now deprecated CA20 data set).

| WRF ensemble member name | Reanalysis product | PBL scheme | SST product | LSM | Surface layer scheme |
|---|---|---|---|---|---|
| WRF1 | ERA5 | MYNN | OSTIA | NOAH | MYNN |
| WRF2 | ERA5 | YSU | OSTIA | NOAH | MM5 |
| WRF3 | ERA5 | MYNN | OSTIA | NOAH-MP | MYNN |
| WRF4 | ERA5 | YSU | OSTIA | NOAH-MP | MM5 |
| WRF5 | ERA5 | MYNN | NCEP | NOAH | MYNN |
| WRF6 | ERA5 | YSU | NCEP | NOAH | MM5 |
| WRF7 | ERA5 | MYNN | NCEP | NOAH-MP | MYNN |
| WRF8 | ERA5 | YSU | NCEP | NOAH-MP | MM5 |
| WRF9 | MERRA-2 | MYNN | OSTIA | NOAH | MYNN |
| WRF10 | MERRA-2 | YSU | OSTIA | NOAH | MM5 |
| WRF11 | MERRA-2 | MYNN | OSTIA | NOAH-MP | MYNN |
| WRF12 | MERRA-2 | YSU | OSTIA | NOAH-MP | MM5 |
| WRF13 | MERRA-2 | MYNN | NCEP | NOAH | MYNN |
| WRF14 | MERRA-2 | YSU | NCEP | NOAH | MM5 |
| WRF15 | MERRA-2 | MYNN | NCEP | NOAH-MP | MYNN |
| WRF16 | MERRA-2 | YSU | NCEP | NOAH-MP | MM5 |

**Table 13.** List of the two WRF ensemble members used for NOW-23 setup selection and validation in the South Pacific region.

| WRF ensemble member name | Reanalysis product | PBL scheme | SST product | LSM | Surface layer scheme |
|---|---|---|---|---|---|
| WRF1 | ERA5 | MYNN | OSTIA | NOAH | MYNN |
| WRF4 | ERA5 | YSU | OSTIA | NOAH-MP | MM5 |

lidars. The setup shows a near-zero bias at all considered heights, which represents a great improvement compared to the bias found in the CA20 data set.

Also, as done for the mid-Atlantic and South Atlantic domains along the U.S. East Coast, we verified that using a 1-month WRF re-initialization period in our simulations does not impact the validation metrics as time goes by in each calendar month (figure not shown).

## 8.5 Choice of the long-term WRF setup

Table 14 details the WRF setup that we select and use for the NOW-23 long-term WRF simulation in the South Pacific region,
which covers the period from January 1, 2000, to December 31, 2022.

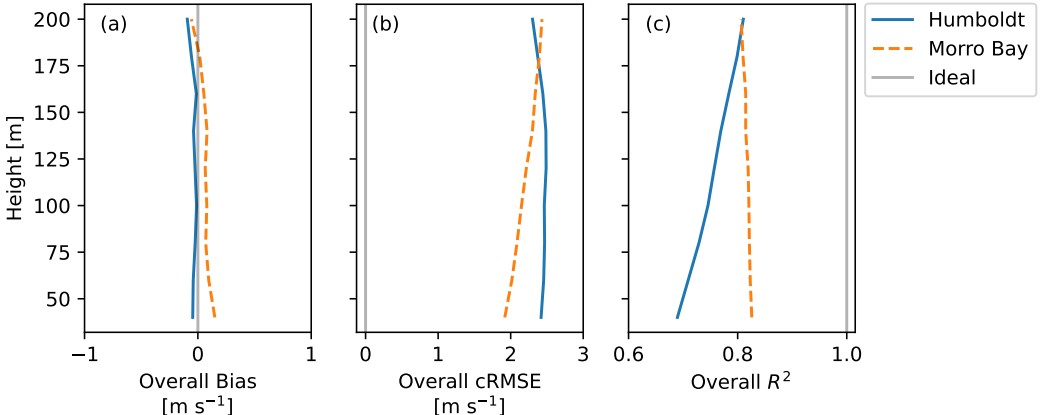

**Figure 16.** Vertical profiles of bias, cRMSE, and r for the WRF4 setup at the locations of the Humboldt and Morro Bay lidars.

**Table 14.** Selected WRF setup for the long-term NOW-23 data set in the South Pacific region.

| WRF ensemble member name | Reanalysis product | PBL scheme | SST product | LSM | Surface layer scheme |
|---|---|---|---|---|---|
| WRF4 | ERA5 | YSU | OSTIA | NOAH | MM5 |

## 8.6 Long-term offshore wind resource

The 23-year mean wind speed at 160 m a.s.l. for the South Pacific region is shown in Fig. 17, and its seasonal and diurnal variabilities are described in Appendix A.

Additionally, we show in Fig. 18 the difference in mean wind speed at 160 m a.s.l. between NOW-23 (2000–2022) and the now-deprecated CA20 data set (2000–2019). We observe that NOW-23 models, on average, lower wind speed across the whole region, with the largest difference (close to 1.5 m s$^{-1}$), in northern California, near the Humboldt wind energy lease area.

## 9 NOW-23 data set in the North Pacific region

No publicly available hub-height offshore wind speed observations exist in the North Pacific domain. Therefore, the WRF setup chosen for this region is based on the results obtained in the early stages of the NOW-23 development, when both the sensitivity analyses in the South Pacific (for the CA20 data set) and mid-Atlantic domains pointed toward using a MYNN-based WRF setup. Thus, this setup is selected in the North Pacific region with no additional validation against observations.

### 9.1 Choice of the long-term WRF setup

Table 15 details the WRF setup that is selected and used for the long-term WRF simulation in the North Pacific region, which covers the period from January 1, 2000, to December 31, 2019.


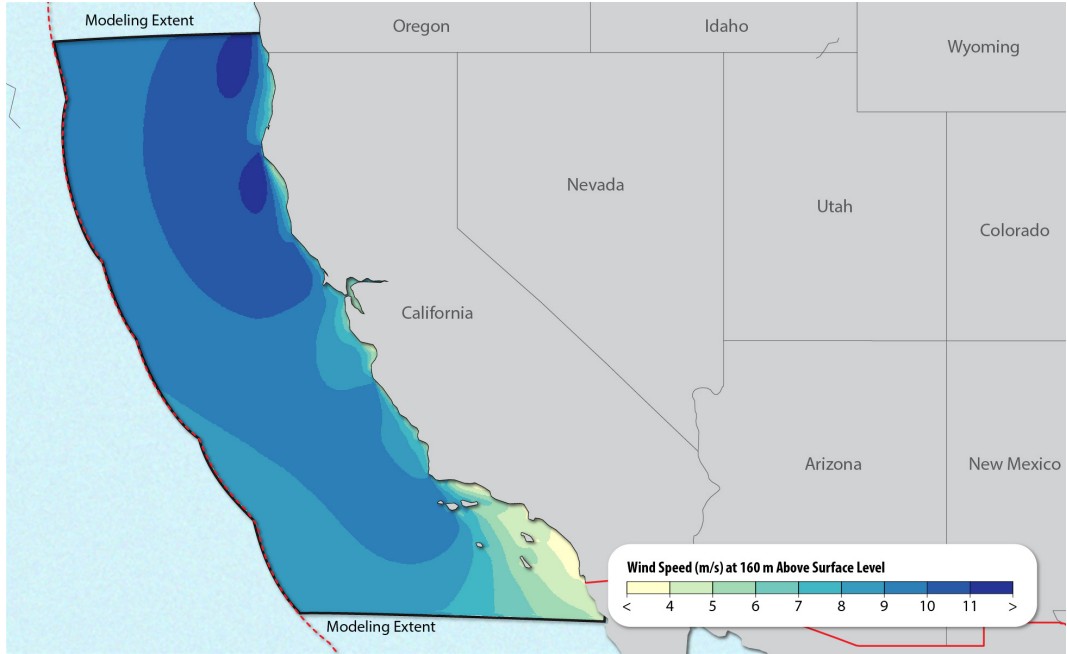

**Figure 17.** Map of the 22-year (2000–2022) mean wind speed at 160 m a.s.l. for the South Pacific region. The red dashed line represents the limit of the U.S. EEZ. The continuous black line, where not overlaid with the EEZ boundary, shows the limit of the NOW-23 WRF domain.

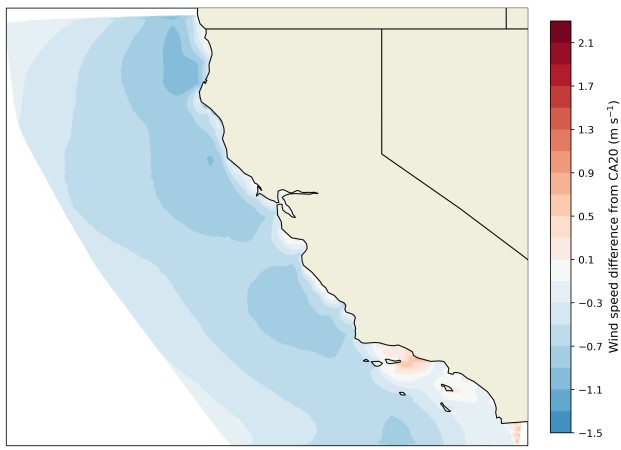

**Figure 18.** Map showing the difference in mean 160-m wind speed between the NOW-23 data set and the CA20 data set.

## 9.2 Long-term offshore wind resource

The 20-year mean wind speed at 160 m a.s.l. for the North Pacific region is shown in Fig. 19.



**Table 15.** Selected WRF setup for the long-term NOW-23 data set in the North Pacific region.

| WRF ensemble member name | Reanalysis product | PBL scheme | SST product | LSM | Surface layer scheme |
|---|---|---|---|---|---|
| WRF1 | ERA5 | MYNN | OSTIA | NOAH | MYNN |

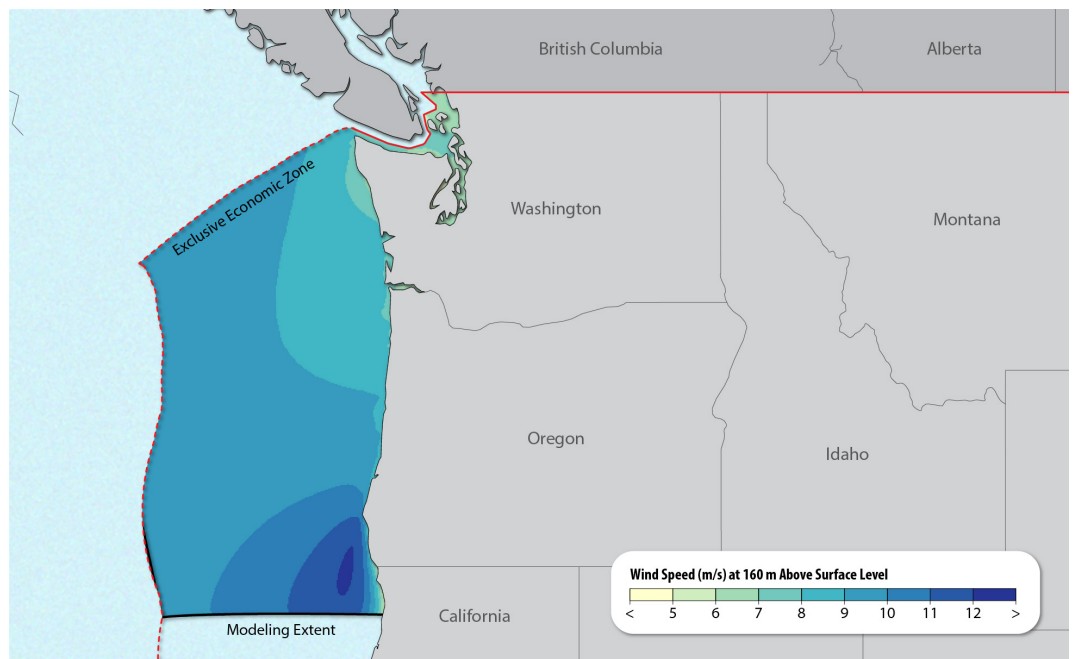

**Figure 19.** Map of the 20-year (2000–2019) mean wind speed at 160 m a.s.l. for the North Pacific region. The red dashed line represents the limit of the U.S. EEZ. The continuous black line, where not overlaid with the EEZ boundary, shows the limit of the NOW-23 WRF domain.

## 9.3 Future work

Because the review of the first-generation CA20 South Pacific data set, generated using the MYNN PBL parameterization, revealed a significant wind speed bias, we extend the review northward through the North Pacific coast. This preliminary examination consists of a comparison between two WRF simulations with different physics configurations, run for year 2020. The first simulation mimics the setup of the main 20-year run for this region, and we compare that with a second simulation that uses the YSU PBL scheme and the MM5 surface layer parameterization. We find that the mean wind speed findings are consistent with those in the South Pacific data set, with stronger wind speeds off the coast resulting from the MYNN/MYNN (PBL scheme/surface layer parameterization) setup compared with that of YSU/MM5. Differences are larger (>0.5 m s$^{-1}$) off the southern coast of Oregon and decrease in magnitude in the northern half of the regional domain and in the open ocean.





**Table 16.** Selected WRF setup for the long-term NOW-23 data set in the Hawaii region.

| WRF ensemble member name | Reanalysis product | PBL scheme | SST product | LSM | Surface layer scheme |
|---|---|---|---|---|---|
| WRF1 | ERA5 | MYNN | OSTIA | NOAH | MYNN |

While no hub-height observations of offshore wind in the region are available, model validation can be performed using near-surface buoy data as well as coastal observations. Leveraging available observations will be essential to assess the accuracy of the chosen WRF setup in the region and will be subject to future work, pending funding availability. In the meantime, NREL
and its project partners warrant caution for the stakeholders interested in using the NOW-23 data set in this region.

## 10  NOW-23 data set in Hawaii

Similar to the North Pacific region, we conduct no validation for the Hawaii domain, where no observations of hub-height offshore wind were publicly available when this regional analysis was initiated at the early stages of the project. The WRF setup used in the NOW-23 Hawaii data set is based on the results obtained in the South Pacific (for its now-deprecated CA20
data set) and the mid-Atlantic domains.

### 10.1  Choice of the long-term WRF setup

Table 16 details the WRF setup that we select and use for the long-term WRF simulation in the Hawaii region, which covers the period from January 1, 2000, to December 31, 2019.

### 10.2  Long-term offshore wind resource

The 20-year mean wind speed at 160 m a.s.l. for the Hawaii region is shown in Fig. 20. We find a larger spatial variability in the mean offshore wind speed compared to the other offshore regions. In general, mean winds are stronger south of the Hawaii islands and on the northern side of the archipelago. A strong wind resource is also observed in the channel between the islands of Hawaii and Maui. Appendix A shows the variability of the long-term wind resource.

### 10.3  Future work

A floating lidar was deployed in the Hawaii region by PNNL in December 2022, and its data are publicly available. The observations collected by this instrument, together with other near-surface and onshore data sets, will be essential in assessing the accuracy of the chosen WRF setup in the region, and will be subject to future work, pending funding availability. In the meantime, NREL warrants caution for the stakeholders interested in using this data set in this region.



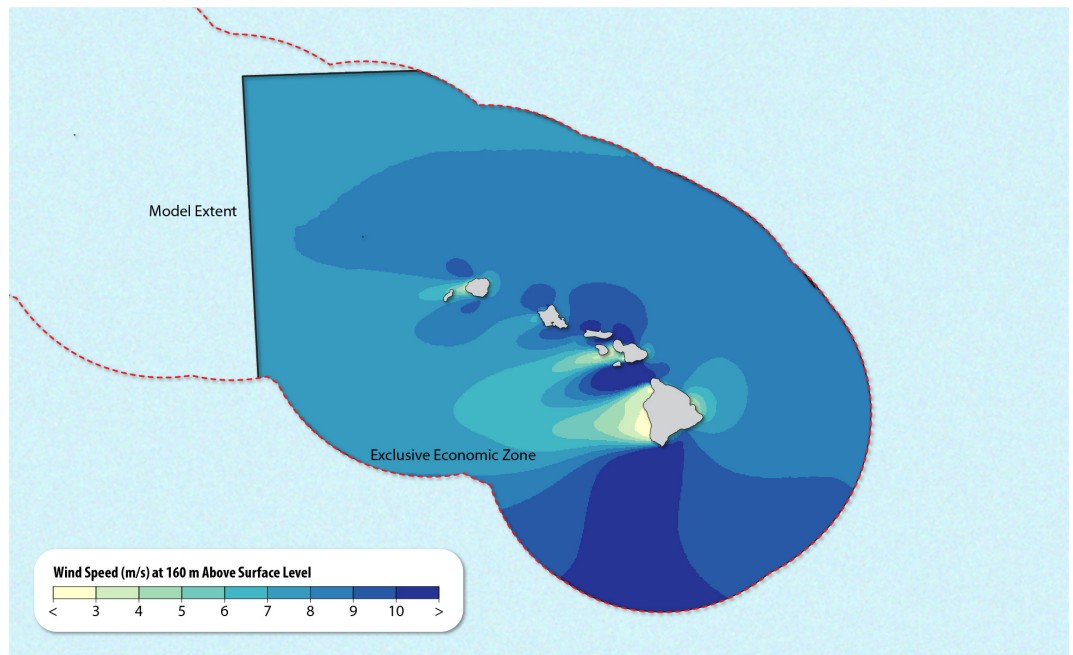

**Figure 20.** Map of the 20-year (2000–2019) mean wind speed at 160 m a.s.l. for the Hawaii region. The red dashed line represents the limit of the U.S. EEZ. The continuous black line, where not overlaid with the EEZ boundary, shows the limit of the NOW-23 WRF domain.

## 11 Uncertainty quantification

NOW-23 is a modeled data set, and, as such, it comes with inherent, unavoidable uncertainty. As part of the NOW-23 development, significant effort was spent to provide stakeholders with this uncertainty information. We tackle this aspect from different points of view, which are briefly summarized in the next paragraphs. For each topic listed below, we refer to other peer-reviewed publications for a detailed description of the analyses performed.

### 11.1 Ensemble-based uncertainty

First, we focus on the characterization of what we call the ensemble-based uncertainty, or boundary condition and parametric uncertainty, in modeled wind speed. When considering NWP models, the choices of the model setup and inputs have a direct impact on the model wind speed prediction and therefore on its uncertainty. Estimating the boundary condition and parametric uncertainty associated with modeled wind speed requires running a model ensemble, which can be computationally intensive for producing long-term data over large regions as for the NOW-23 data set. Therefore, we propose two alternative approaches

that use a short-term mesoscale ensemble (i.e., the 1-year WRF ensemble we use for model setup selection in several offshore regions) alongside a single model run for the desired long-term period (i.e., the 20+ year NOW-23 run). We quantify hub-height wind speed boundary condition and parametric uncertainty from the short-term model ensemble as its normalized across-ensemble standard deviation. Then, we use a gradient-boosting model and an analog ensemble approach to extrapolate



the uncertainty to the full 20+ year period.


We test our proposed methods in the South Pacific domain (using the now deprecated MYNN-based CA20 data set) and find that both approaches provide accurate estimates of long-term wind speed boundary condition and parametric uncertainty ($r^2 > 0.75$), with the gradient-boosting model performing slightly better than the analog ensemble. We also assess the physical variability in the uncertainty estimates and find that wind speed uncertainty increases closer to land, stable and unstable cases have larger uncertainty than neutral conditions, and winter has a smaller boundary condition and parametric sensitivity than summer. Finally, we report a median hourly uncertainty between 10% and 14% of the mean 100 m wind speed values across the offshore wind energy lease areas in the region. The results of this analysis are described in detail in Bodini et al. (2021).

### 11.2 Model uncertainty compared to observations

While helpful from a modeling point of view, the assessment of the boundary condition and parametric uncertainty presents several limitations. In fact, the magnitude of this ensemble-based uncertainty is strictly connected to the (limited) number of choices sampled within the considered model setups, so that only a limited component of the actual wind speed error with respect to observations (our best proxy for the true wind speed) can be quantified from it. The full uncertainty in NWP-model-predicted wind speed can be quantified only when direct observations of the wind resource are available. In this scenario, the residuals between modeled and observed wind speed can be calculated, and the model error is quantified in terms of its bias (i.e., the mean of the residuals) and uncertainty (i.e., the standard deviation of the residuals). The obtained model uncertainty would then be added to the inherent uncertainty of the wind speed measurements.

We apply this approach in the mid-Atlantic region. Given the lack of long-term (20+ year) hub-height offshore wind speed observations in the region (the same applies to all the U.S. offshore regions), we propose a methodological framework to leverage both floating lidar and near-surface buoy observations to quantify uncertainty in the long-term modeled hub-height wind resource. We train and validate a machine learning technique to vertically extrapolate near-surface wind speed to hub height using the available short-term lidar data sets in the region. We then apply this model to vertically extrapolate the long-term near-surface buoy wind speed observations to hub height for comparison to the long-term NOW-23 data set. Using this comprehensive approach, we find that the mean 20-year uncertainty (including the uncertainty coming from the observations and from the application of the machine learning approach) in 140 m wind speed is slightly lower than 3 m s$^{-1}$ across the considered region, with larger uncertainty in stable conditions. The results of this analysis are described in detail in Bodini et al. (2023).

## 12 NOW-WAKES: a post-construction data set for the mid-Atlantic wind energy areas

A promising offshore wind resource is often located near large population centers, so that a rapid wind plant development is expected. However, wind turbines and wind plants generate wakes, which are regions of reduced wind speed that may





negatively impact downwind turbines and plants. As part of the NOW-23 data set, we developed NOW-WAKES, a 1-year post-construction data set to model and assess the impact of offshore wakes from the upcoming wind plants in some of the lease and call areas (as of 2019) in the mid-Atlantic region. As part of the NOW-WAKES effort, we also investigate the uncertainty connected to some of the modeling choices made in this 1-year analysis by using two different PBL schemes for these simula-
tions. Here, we report a brief summary of the two studies and refer to two peer-reviewed journal articles we published on the topic for further details.

To assess wake variability and annual energy production, we conduct a year-long series of simulations with no wind plants, one wind plant, and complete build-out of the lease areas using WRF and its Fitch wind farm parameterization (Fitch et al.,
2012) to calculate wake effects and distinguish between wakes generated within one plant and those generated externally between plants. Wind plant simulations sited 12 MW turbines at a density of 3.14 MW km$^{-2}$, matching the planned density of 3 MW km$^{-2}$. The strongest wakes, propagating 55 km, occur during stable stratification in summer, which coincides with peak grid demand in New England. The offshore region experiences much stronger seasonal variability in wakes compared to diurnal variability. The mean year-long wake impacts reduce power output by roughly 35%, with internal wakes causing
greater power losses (27% on average) than external wakes (14% on average). The results of this analysis are described in detail in Rosencrans et al. (2023).

While the results of the analysis above use a state-of-the-art approach for wake modeling in WRF, different simulation methodologies can be used to estimate the impact of wind plant wakes, each with their own level of accuracy and sensitivity
to input parameters. For example, as seen in the development of the NOW-23 data set, mesoscale simulations without turbines can have variations in hub-height wind speeds due to the choice of PBL scheme. However, the sensitivity of modeled wind plant wakes to different PBL schemes was not explored in the past because wake parameterizations were only compatible with the MYNN PBL scheme. To address this limitation, we couple the WRF Fitch wind farm parameterization (Fitch et al., 2012) with the new NCAR 3DPBL scheme (Kosović et al., 2020; Juliano et al., 2022) and compare the results to those obtained using
the MYNN PBL scheme. We simulate an idealized wind plant under stable, neutral, and unstable conditions with matching hub-height wind speeds using the two PBL schemes. The average losses in hub-height wind speeds within the plant differ between the two schemes by up to $-0.20$ to $0.22$ m s$^{-1}$, and correspondingly, capacity factors range from 39.5% to 53.8%. These results suggest that the choice of the PBL scheme can also contribute to uncertainty in modeled wakes, and therefore we recommend including PBL variability in wind plant planning sensitivity and forecasting studies. The results of this analysis
are described in detail in Rybchuk et al. (2022).





## 13 Code and data availability

The whole NOW-23 data set is publicly available at no cost through Amazon Web Services (AWS) at https://doi.org/10.25984/1821404 (Bodini et al., 2020). The data are stored as both HDF5 and WRG files. The same page also includes a link to the NOW-23 WRF namelists.

The HDF5 files are available for each offshore region at both 5-minute resolution and as hourly averages. The following variables are available in the HDF5 files:

- Wind speed at 10 m, 20 m intervals between 20 m and 300 m, 400 m, and 500 m (m s$^{-1}$)

- Wind direction at 10 m, 20 m intervals between 20 m and 300 m, 400 m, and 500 m (° from N)

- Planetary boundary layer height (m)

- Pressure at 0 m, 100 m, 200 m, and 300 m (Pa)

- Temperature at 2 m, 10 m, 20 m intervals between 20 m and 300 m, 400 m, and 500 m (°C)

- Friction velocity at 2 m (m s$^{-1}$)

- Surface heat flux (W m$^{-2}$)

- Inverse Monin–Obukhov length at 2 m (m$^{-1}$)

- Sea surface temperature (°C)

- Skin temperature (°C)

- Relative humidity at 2 m (%)

- Roughness length (m)

To facilitate accessing, extracting, and manipulating data from the NOW-23 large files, NREL has developed the Resource 560 eXtraction (rex) tool. Instructions on how to install rex, as well as examples of its usage, can be found at nrel.github.io/rex. We note that rex allows for, among others, extraction of a subset of variables over a single location, a customized region, and automatic interpolation of the wind speed data to an arbitrary height. In the coming months, the Wind Resource Data Base, a more user-friendly interface to visualize and download the NOW-23 (and other wind resource assessment) data will be released.

The WRG file format is an industry-standard format for publishing modeled wind resource data sets. In general, WRG files provide wind rose information at a specific height at each modeled grid cell. This information is specified in a WRG by providing Weibull fit parameters for reported wind speeds in a given sector. Considerable information is lost when pivoting to a WRG from the HDF5 format. Specifically, all temporal information (e.g., seasonal and diurnal trends) and other relevant



**Table 17.** Data availability for the observational data sets used for NOW-23 validation.

| Data set | Publicly available? | Link |
|---|---|---|
| NDBC buoys | Yes | https://www.ndbc.noaa.gov |
| NYSERDA E05, E06 floating lidars | Yes | https://oswbuoysny.resourcepanorama.dnvgl.com |
| Atlantic Shores floating lidar | Yes | https://erddap.maracoos.org/erddap/tabledap/ |
| UMaine floating lidar | Mostly not | Mean data available in Viselli et al. (2022) |
| DOE Virginia floating lidar | Yes | https://a2e.energy.gov/ds/buoy/buoy.z01.a0 |
| Michigan Mid-Plateau lidar | No | N/A |
| Shell Gulf of Mexico floating lidars | No | N/A |
| Morro Bay floating lidar | Yes | https://a2e.energy.gov/ds/buoy/lidar.z06.b0 |
| Humboldt floating lidar | Yes | https://a2e.energy.gov/ds/buoy/lidar.z05.b0 |
| NOAA coastal wind profilers | Yes | https://psl.noaa.gov/data/obs/data/ |

atmospheric parameters are discarded, with only wind speed and direction being preserved. Regardless, the WRG format has

been the industry standard for decades and is still actively used for offshore wind resource assessment. For the NOW-23 data sets, NREL has provided WRG files calculated at 160 m and at 30° wind direction sector bins. An open-source codebase has also been publicly released (github.com/NREL/wrg_maker) so that stakeholders can produce additional WRG files from the NOW-23 data set, if desired.

To further facilitate access to the NOW-23 data, Bodini et al. (2020) also includes long-term data (in easily accessible .csv format) at the locations of the lidars used to validate the NOW-23 data set (locations in Figure 2).

Any request for technical support on the NOW-23 data set can be directed to windtoolkit@nrel.gov.

Some of the observations used for model validation are also publicly available. Details about the observational data sets used

are included in Table 17.

## 14    Conclusions

The NOW-23 data set is a cutting-edge, offshore-focused wind resource assessment data set that aims to support the growth of the offshore wind energy sector in the United States. This comprehensive overview paper outlines the approach employed to generate and validate the data set for each U.S. offshore region (with the exception of Alaska). The national mean long-term

wind speed, depicted in Fig. 21, is derived by amalgamating all the regional domains.

Anticipated to replace NREL's WIND Toolkit and become one of the most widely utilized data sets for offshore wind resource assessment in the United States, the NOW-23 data set showcases the significance of conducting regionally focused validation and selecting appropriate model setups based on multiple observational data sets. The analysis underscores the sym-

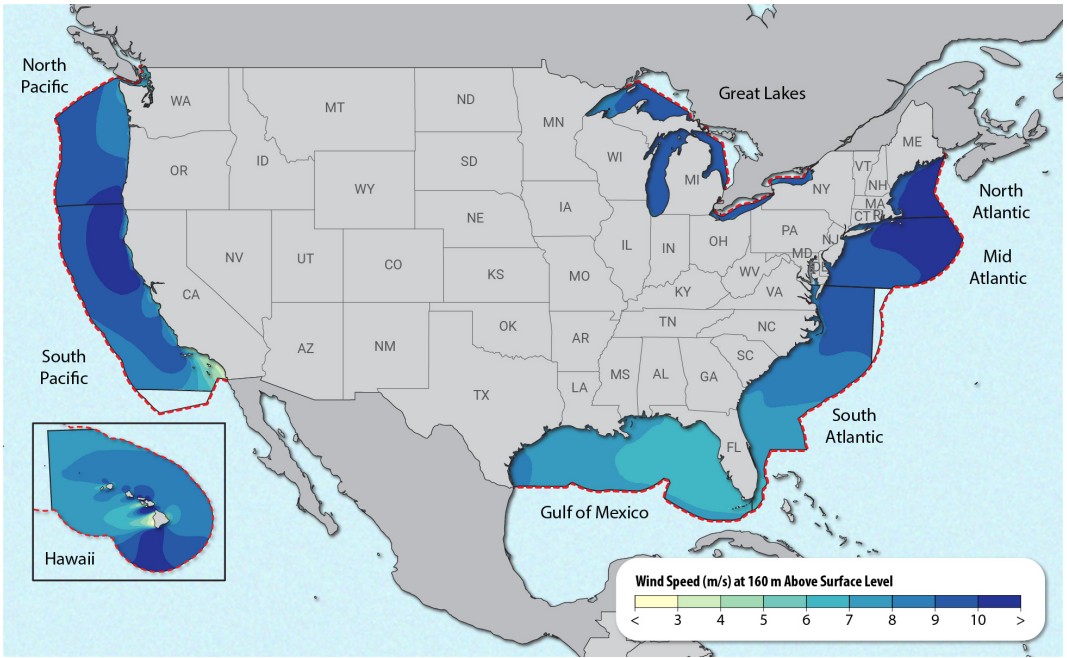

**Figure 21.** Mean long-term wind speed at 160 m from the NOW-23 data set, for all U.S. offshore regions (except for Alaska). The red dashed lines represent the limit of the U.S. EEZ. The continuous black lines show where data from the regional domains were cut to create this national-scale map.

biotic relationship between NWP models and observations, emphasizing their interconnectedness. To mitigate the inherent
uncertainty in numerical models, an increased quantity of long-term observations is required, which can be facilitated through the sharing of proprietary observational data sets. In assessing the costs and benefits of data-sharing initiatives, it is essential for stakeholders to consider the long-term advantages that access to additional observational data sets can offer in terms of enhanced numerical modeling.

Further analysis and validation are needed for the NOW-23 data set in Hawaii and the North Pacific, subject to future work contingent on funding availability. As more offshore observations become accessible, the current validation efforts can be expanded to ensure the NOW-23 model setup is validated in regions of interest for present and future offshore wind energy lease areas. Additionally, regular updates to the NOW-23 data set (pending funding availability) are desirable, enabling stakeholders to employ the industry-standard measure-correlate-predict approach as they collect short-term observations in the future. Also,
an analysis similar to what done to create NOW-WAKES can be replicated in different offshore regions. As offshore wind turbines are built in the U.S., the simulated wake effects can and should be validated against any available observations. Lastly, given the unsatisfactory performance of the MYNN PBL scheme on the U.S. West Coast, further analysis is warranted to inves-

tigate the causes of its failure in the region and propose and implement offshore-focused improvements to the parameterization scheme.

**Appendix A: Seasonal and diurnal variability of long-term offshore wind speed in the NOW-23 data set**

In all offshore regions modeled in NOW-23, we observe distinct seasonal and diurnal patterns in hub-height wind speed.

On the U.S. East Coast, we observe consistent seasonal and diurnal cycles in hub-height wind speed. During the winter months, winds are at their strongest, gradually weakening as we move into summer. The seasonal differences in wind speed

become more pronounced further offshore. A comparison of the month of January, which typically experiences the highest average wind speed, with August, when wind speed is at its lowest, reveals differences sometimes exceeding 6 m s$^{-1}$ (Figs. A1, A3, and A5). Additionally, a diurnal cycle is evident, with variations closer to the shore being more pronounced, likely because of sea breeze effects. In the evenings, we observe stronger winds, while the morning hours tend to have lower wind speeds (Figs. A2, A4, and A6).


We find similar results in the Gulf region, where winter months have the strongest winds, and summer months have the lowest wind speeds. The area with the largest deviation from the annual mean varies throughout the year, with the eastern side of the Gulf experiencing the largest deviation in early winter and the western side experiencing it later in the season. A similar transition occurs from east to west during the summer months (Fig. A7). On a diurnal basis, we observe stronger winds at night

and lower winds during the day, with the most significant deviations once again near the coast (Fig. A8).

When considering the Great Lakes region, we observe the same seasonal and diurnal cycles in hub-height wind speed as on the U.S. East Coast: the strongest winds occur in winter and early spring, while the lowest winds occur in summer, with greater seasonal differences in lakes more to the south (Fig. A9). The diurnal cycle in this region is slightly delayed compared to the

U.S. East Coast, with stronger winds in the late evening and lower winds around midnight (Fig. A10).

In the North Pacific, we find consistent seasonal and diurnal cycles in hub-height wind speed. Once again, winter months have the strongest winds, while summer months have the lowest winds, except for the southern part of the ocean west of the Oregon coast, where the annual cycle is opposite (Fig. A11). On a diurnal basis, we observe stronger winds in the afternoon

and early night, and lower winds in the late nights and early mornings, with the largest deviations near the coast (Fig. A12).

In the South Pacific, a different annual cycle emerges (Fig. A13), with strongest winds observed in the spring and early summer, with the southern portion of the domain experiencing stronger winds earlier than northern California. On a diurnal basis, we find similar results to what was observed in the adjacent North Pacific region, with stronger winds in the afternoon

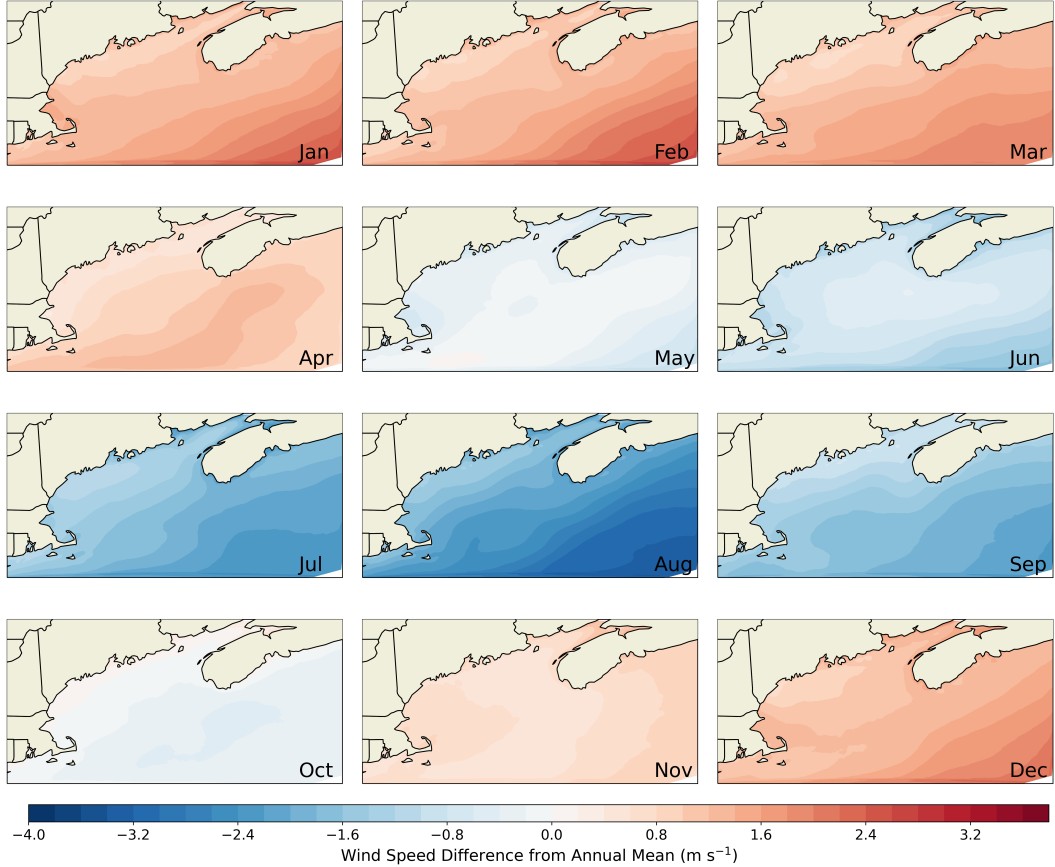

**Figure A1.** Maps showing the annual cycle in 160 m wind speed, expressed in terms of the difference for each month's mean wind speed from the overall mean, for the North Atlantic region.

and early night (Fig. A14).

For Hawaii, the seasonal and diurnal cycles in hub-height wind speed are not as clear as in the other offshore regions. Winds close to the islands are stronger in the summer and weaker in the winter. Further offshore, the seasonal variability becomes less clear (Fig. A11). On a diurnal basis, we find stronger winds closer to the islands in the afternoon and early night, and lower
winds in the late night and early morning, with the largest deviations near the coast (Fig. A16).

**Appendix B: Wind speed differences between NOW-23 overlapping regional domain boundaries**

When two NOW-23 WRF domains are adjacent (North and mid-Atlantic, mid-Atlantic and South Atlantic, South Atlantic and Gulf of Mexico, and North and South Pacific), there is some limited spatial overlap. In this section, we compare the mean 160-m wind speed from neighboring domains in these limited overlapping regions to quantify the deviation between regional

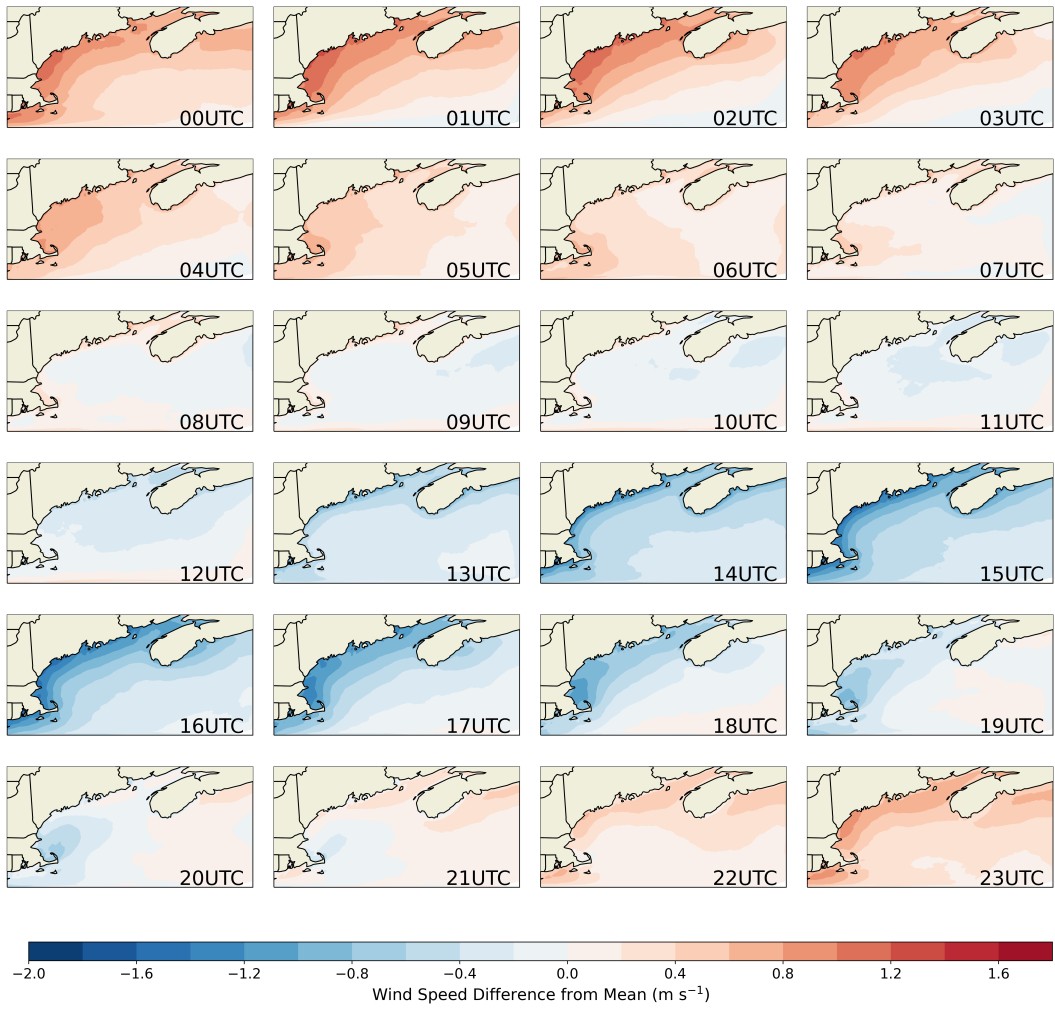

**Figure A2.** Maps showing the diurnal cycle in 160 m wind speed, expressed in terms of the difference for each UTC hour's mean wind speed from the overall mean, for the North Atlantic region.

data sets and provide guidance to NOW-23 stakeholders.

In most cases, we find minimal differences between the NOW-23 regional data sets when they overlap, consistently well within the model uncertainty. The deviation between the overlapping area from the mid-Atlantic and the North Atlantic regional data sets (Fig. B1), which both use the same WRF setup (with the MYNN PBL scheme), is smaller than 0.2 m s$^{-1}$

in either direction. Similarly, we observe limited differences in the overlapping region between the South Atlantic and the mid-Atlantic domains (Fig. B2), which employ different WRF setups. Near the coast, the difference is smaller than 0.2 m s$^{-1}$ in either direction. The mean difference increases in the open ocean, in areas not directly relevant for offshore wind energy

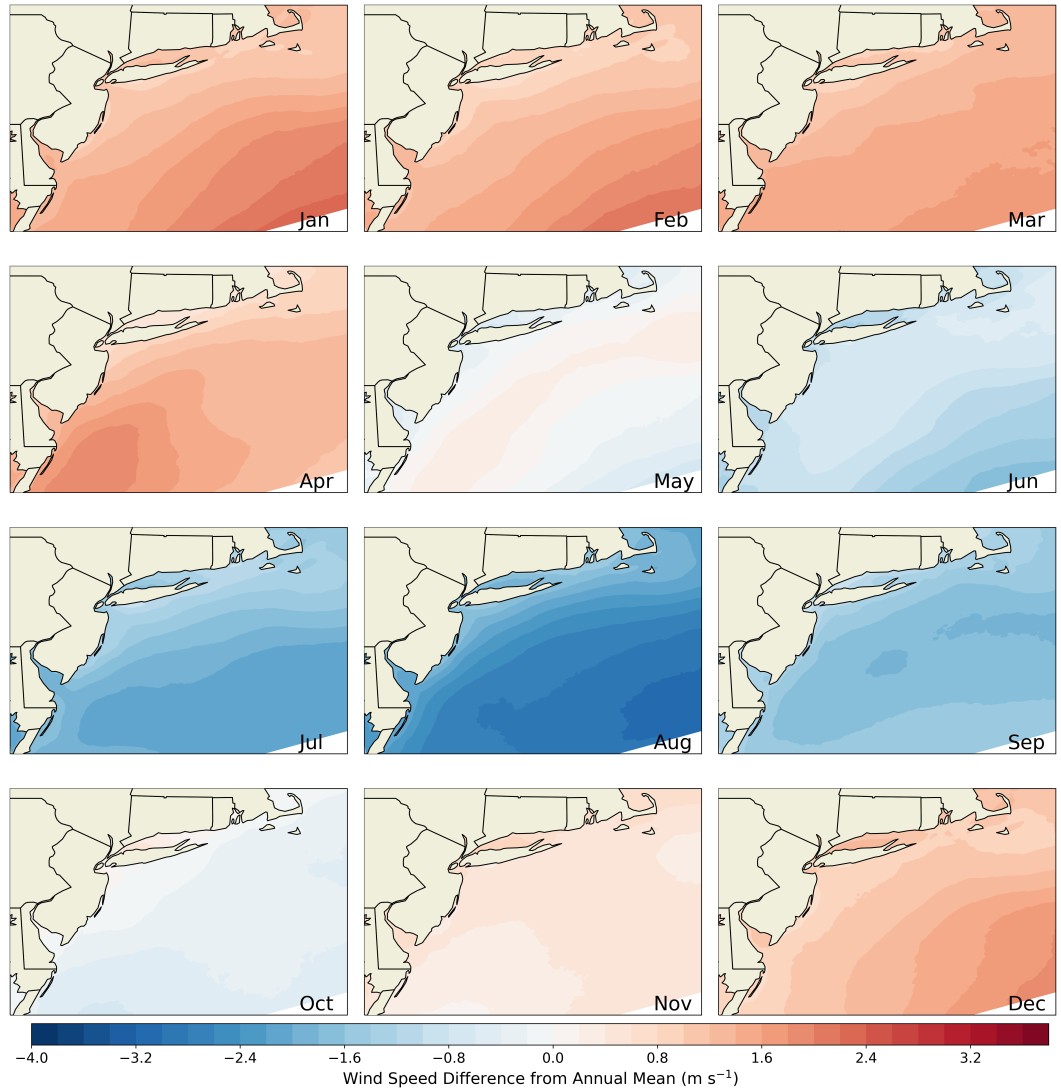

**Figure A3.** Maps showing the annual cycle in 160 m wind speed, expressed in terms of the difference for each month's mean wind speed from the overall mean, for the mid-Atlantic region.

purposes. Additionally, the South Atlantic data set slightly (<0.2 m s$^{-1}$) models stronger wind speed compared to the Gulf of Mexico domain (Fig. B3), with some larger differences near the south edge of Florida.


On the U.S. West Coast, a major difference appears, with the North Pacific data set modeling significantly stronger wind speed compared to the South Pacific one (Fig. B4). This difference is mainly due to the fact the two regional data sets adopt different PBL schemes, which have been shown to produce significantly different results on the U.S. West Coast, as detailed in

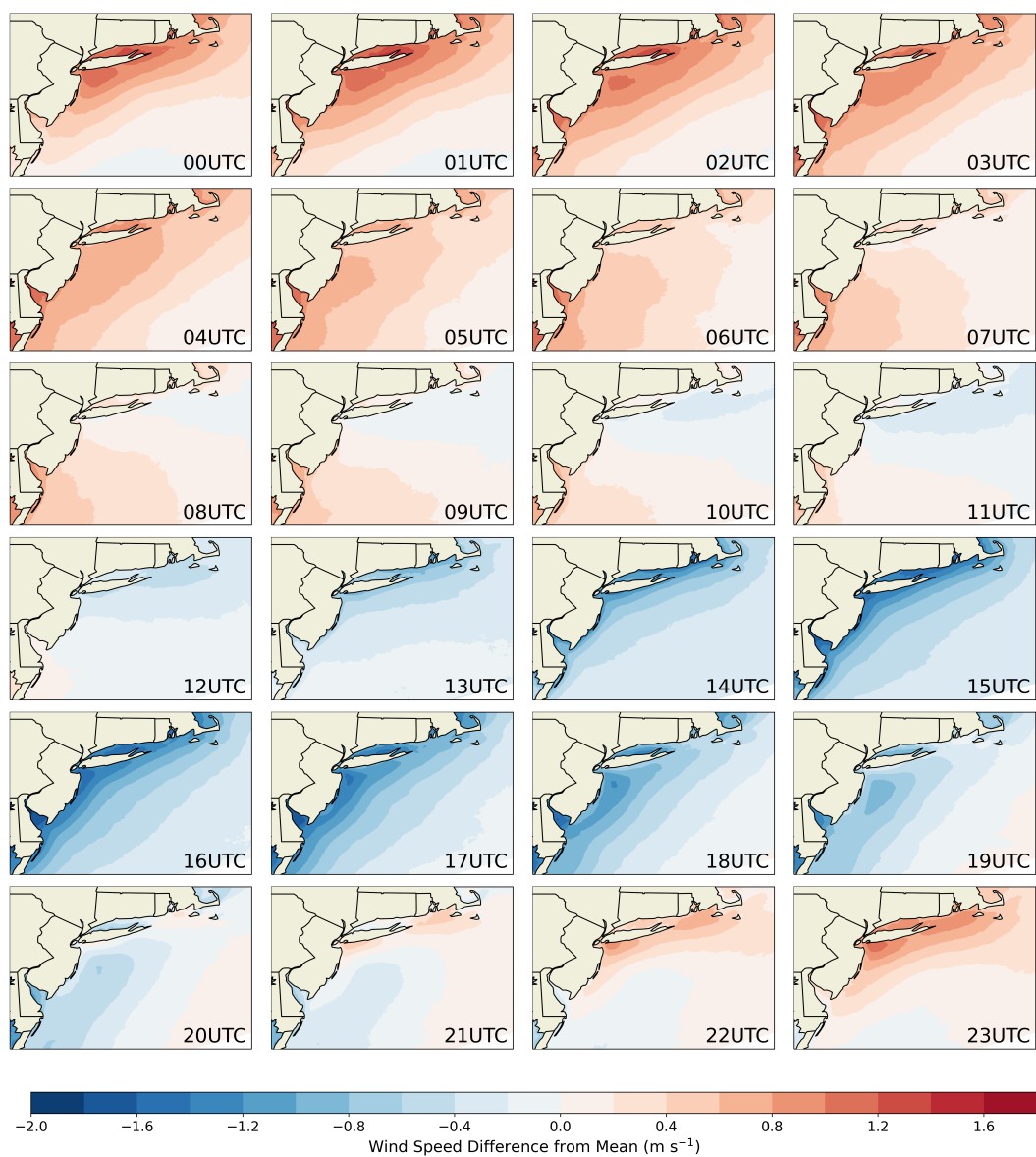

**Figure A4.** Maps showing the diurnal cycle in 160 m wind speed, expressed in terms of the difference for each UTC hour's mean wind speed from the overall mean, for the mid-Atlantic region.

the previous sections.


For these limited areas where the WRF domains overlap, stakeholders can access NOW-23 data from both neighboring regions for download. For all cases where mean differences are limited, the data from either region can be used with confidence. Considering the overlap between the South Atlantic and Gulf of Mexico domains, it is reasonable to expect that the Gulf of

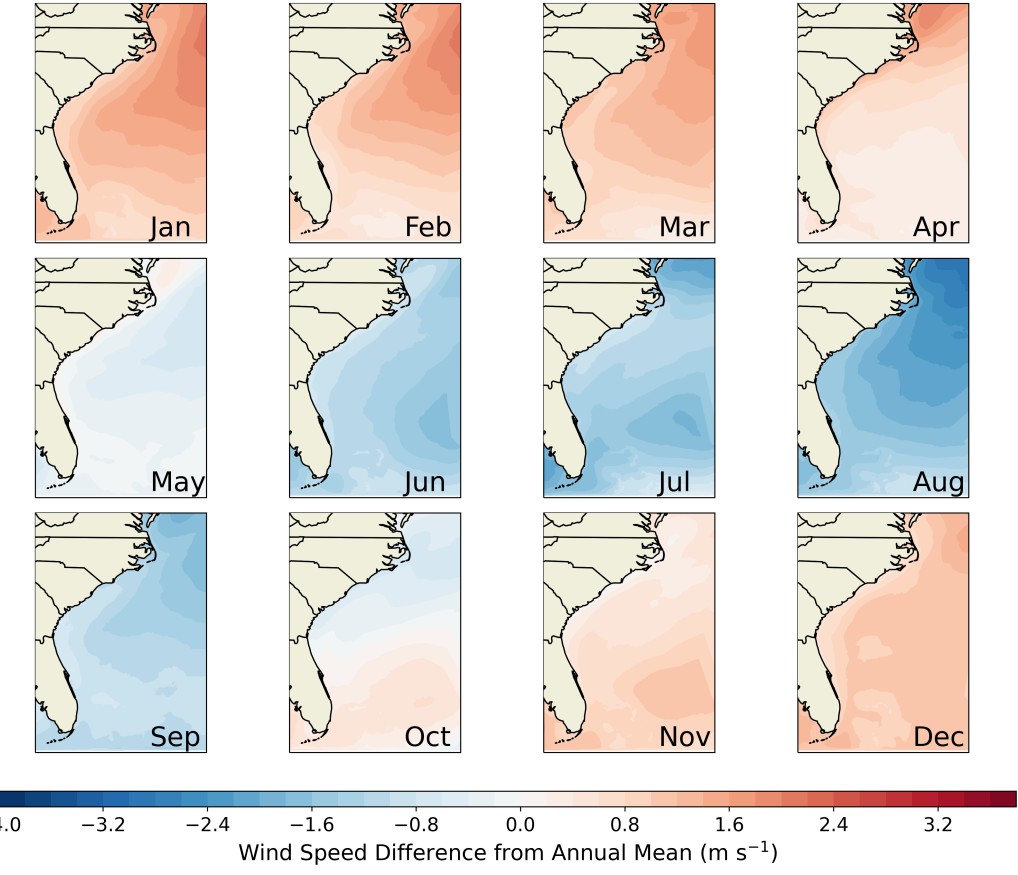

**Figure A5.** Maps showing the annual cycle in 160 m wind speed, expressed in terms of the difference for each month's mean wind speed from the overall mean, for the South Atlantic region.

Mexico regional data set provides the most accurate data in the overlapping area south of Florida, as it captures all the metocean
dynamics in the rest of the Gulf. Similarly, the South Atlantic domain is expected to be more accurate in the small region north of Florida where its WRF domain overlaps with the Gulf of Mexico. Cape Cod could also be considered as a natural barrier influencing somewhat different metocean conditions to the north and south, suggesting that the North Atlantic data set may be better suited for the region north of the Cape, while the mid-Atlantic domain could be preferred for the region south of the Cape. It is important to note, however, that we did not validate these scientific speculations by comparing the NOW-23
modeled data in the overlapping regions against observations, given the limited differences between the regional data sets. For the overlap between the North and South Pacific domains, we would like to remind users that only the South Pacific domain (and its WRF configuration) underwent a regional tuning and validation, and therefore that setup should be preferred for the region of overlap.

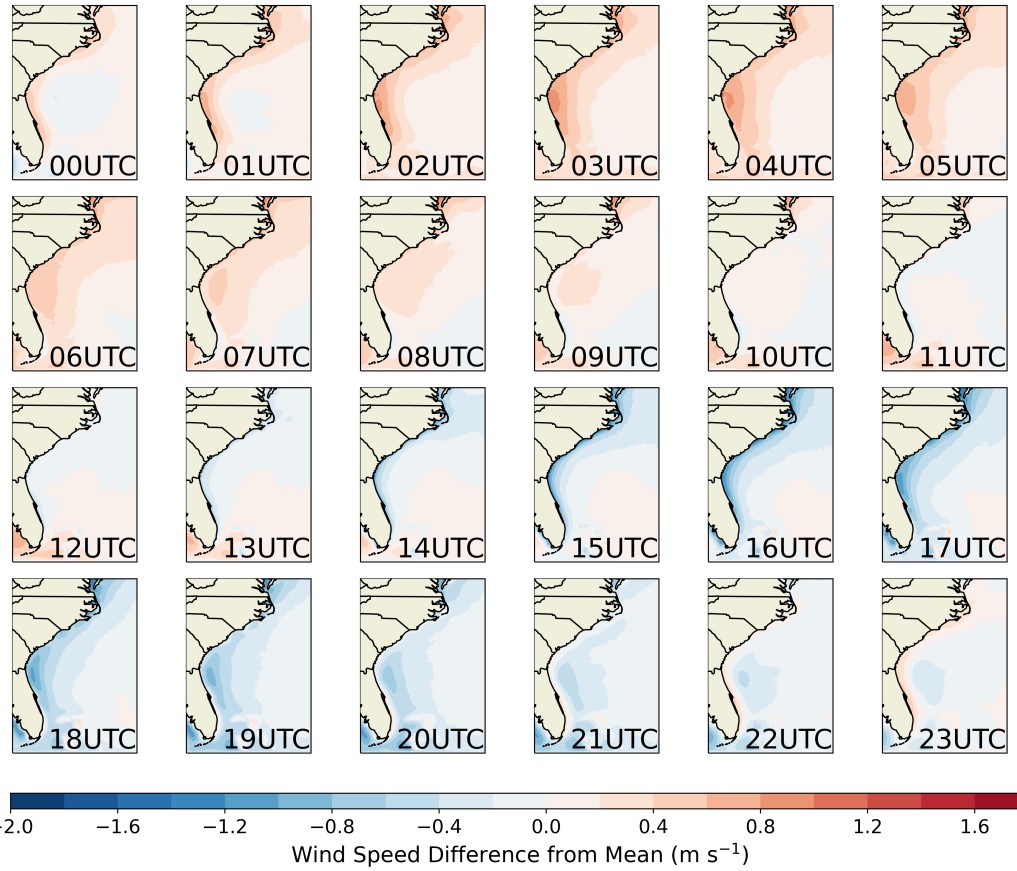

**Figure A6.** Maps showing the diurnal cycle in 160 m wind speed, expressed in terms of the difference for each UTC hour's mean wind speed from the overall mean, for the South Atlantic region.

## Appendix C: Comparison between NOW-23 and the WIND Toolkit

Figure C1 presents a comparison of mean wind speeds between the NOW-23 data set and the 7-year (2007–2013) WIND Toolkit for the offshore regions in the contiguous United States. We note that Hawaii was not part of the WIND Toolkit domain, so a direct comparison between the NOW-23 data set and the WIND Toolkit for this region is not possible. NOW-23 and the WIND Toolkit have many differences (e.g., different length of the period of record, different reanalysis forcing, different WRF version). However, the magnitude of the differences in mean wind speed between the two data sets seems largely connected, in most regions, to the PBL schemes being adopted in the two data sets. The WIND Toolkit used the YSU PBL scheme across all regions, whereas, as described in this paper, NOW-23 uses either the MYNN or the YSU PBL scheme based on the results of a region-specific validation. For the mid-Atlantic, North Atlantic, Great Lakes, and North Pacific regions, the NOW-23 data set (with the MYNN PBL scheme) consistently models stronger wind speeds compared to the WIND Toolkit, with differences larger than 0.5 m s$^{-1}$ in the mid-Atlantic and North Pacific domains, and even larger in the North Atlantic and Great Lakes


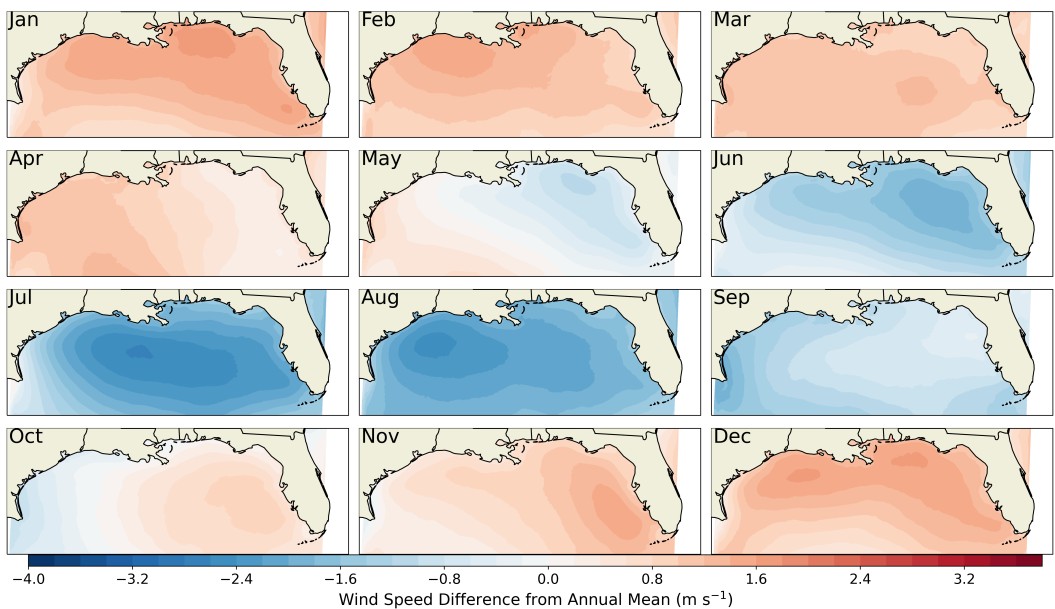

**Figure A7.** Maps showing the annual cycle in 160 m wind speed, expressed in terms of the difference for each month's mean wind speed from the overall mean, for the Gulf of Mexico region.

regions. In the South Atlantic and Gulf of Mexico regions, the wind speeds between the NOW-23 data set (with the YSU PBL scheme) and the WIND Toolkit are similar across the majority of the domains. We note how off the coast of Florida the NOW-23 data set exhibits stronger hub-height wind speeds, with differences larger than $0.5$ m s$^{-1}$ compared to the WIND Toolkit. For the South Pacific region, despite both NOW-23 and the WIND Toolkit using the same PBL scheme (YSU), NOW-23 models stronger hub-height winds off the coast of central and southern California, with some local differences on the order of 1 m s$^{-1}$.

*Author contributions.* Nicola Bodini: Conceptualization, Methodology, Formal analysis, Writing - Original Draft, Supervision, Project administration. Mike Optis: Conceptualization, Methodology, Formal analysis, Writing - Review & Editing, Supervision, Funding acquisition, Project administration. Stephanie Redfern: Methodology, Formal analysis, Writing - Review & Editing. David Rosencrans: Formal analysis, Writing - Review & Editing. Alex Rybchuk: Formal analysis, Writing - Review & Editing. Julie K. Lundquist: Conceptualization, Methodology, Writing - Review & Editing, Supervision. Vincent Pronk: Formal analysis, Writing - Review & Editing. Simon Castagneri: Formal analysis, Writing - Review & Editing. Avi Purkayastha: Software, Writing - Review & Editing. Caroline Draxl: Methodology, Writing - Review & Editing. Ethan Young: Software, Writing - Review & Editing. Billy Roberts: Visualization, Writing - Review & Editing. Evan Rosenlieb: Visualization, Writing - Review & Editing. Walter Musial: Conceptualization, Writing - Review & Editing.

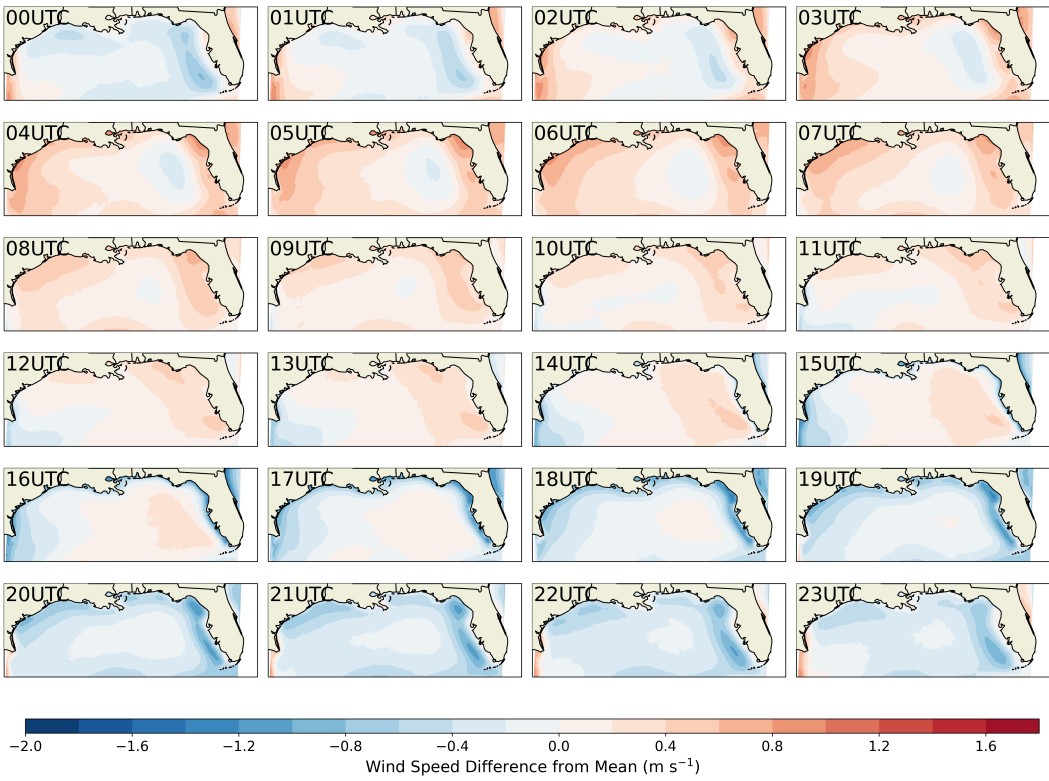

**Figure A8.** Maps showing the diurnal cycle in 160 m wind speed, expressed in terms of the difference for each UTC hour's mean wind speed from the overall mean, for the Gulf of Mexico region.

*Competing interests.* Author Mike Optis co-authored the submitted manuscript while an employee of the National Renewable Energy Laboratory. He has since founded Veer Renewables, which recently released a wind modeling product, WakeMap, which is based on a similar numerical weather prediction modeling framework as the one described in this manuscript. Data from WakeMap is sold to wind energy stakeholders for profit. Public content on WakeMap include a website (https://veer.eco/wakemap), a white paper (https://veer.eco/wp-content/uploads/2023/02/WakeMap_White_Paper_Veer_Renewables.pdf), and several LinkedIn posts promoting WakeMap.

*Acknowledgements.* The authors would like to thank the NOWRDC program managers and the members of the project advisory board for their continuous feedback and support throughout the project. We would like to extend a special thanks to Shell for agreeing to share lidar data in the Gulf of Mexico. This research was performed using computational resources sponsored by the U.S. Department of Energy's Office of Energy Efficiency and Renewable Energy and located at the National Renewable Energy Laboratory.



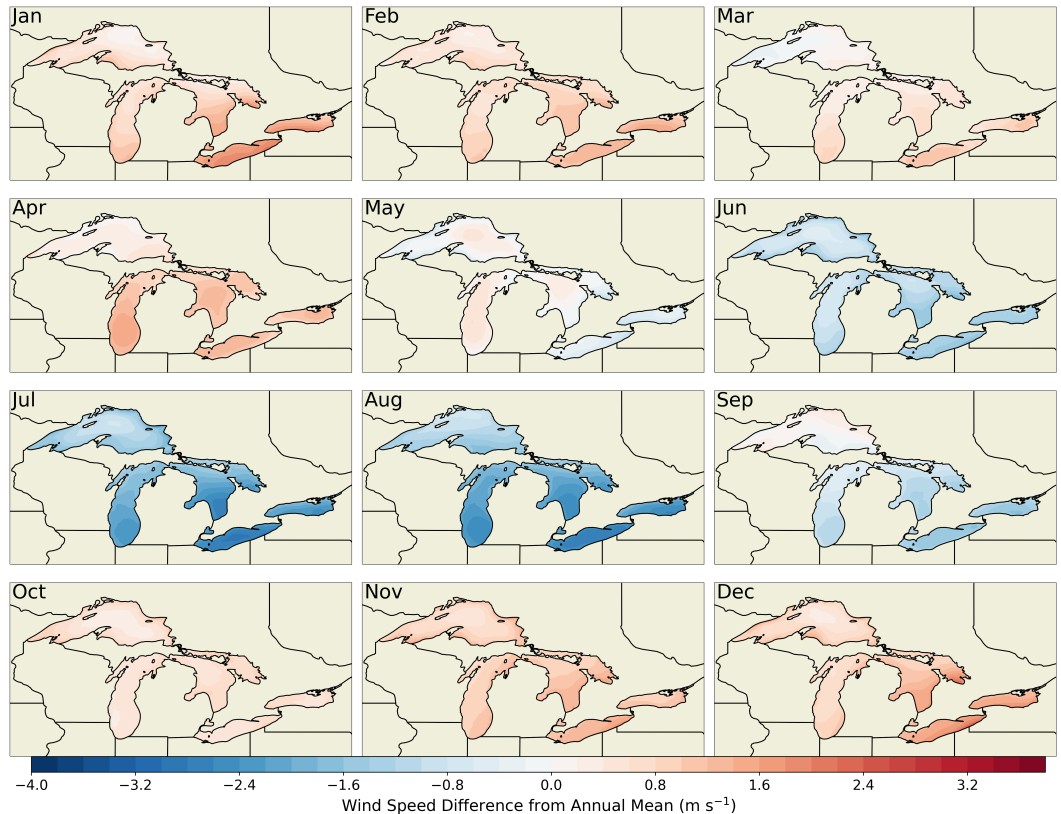

**Figure A9.** Maps showing the annual cycle in 160 m wind speed, expressed in terms of the difference for each month's mean wind speed from the overall mean, for the Great Lakes region.

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



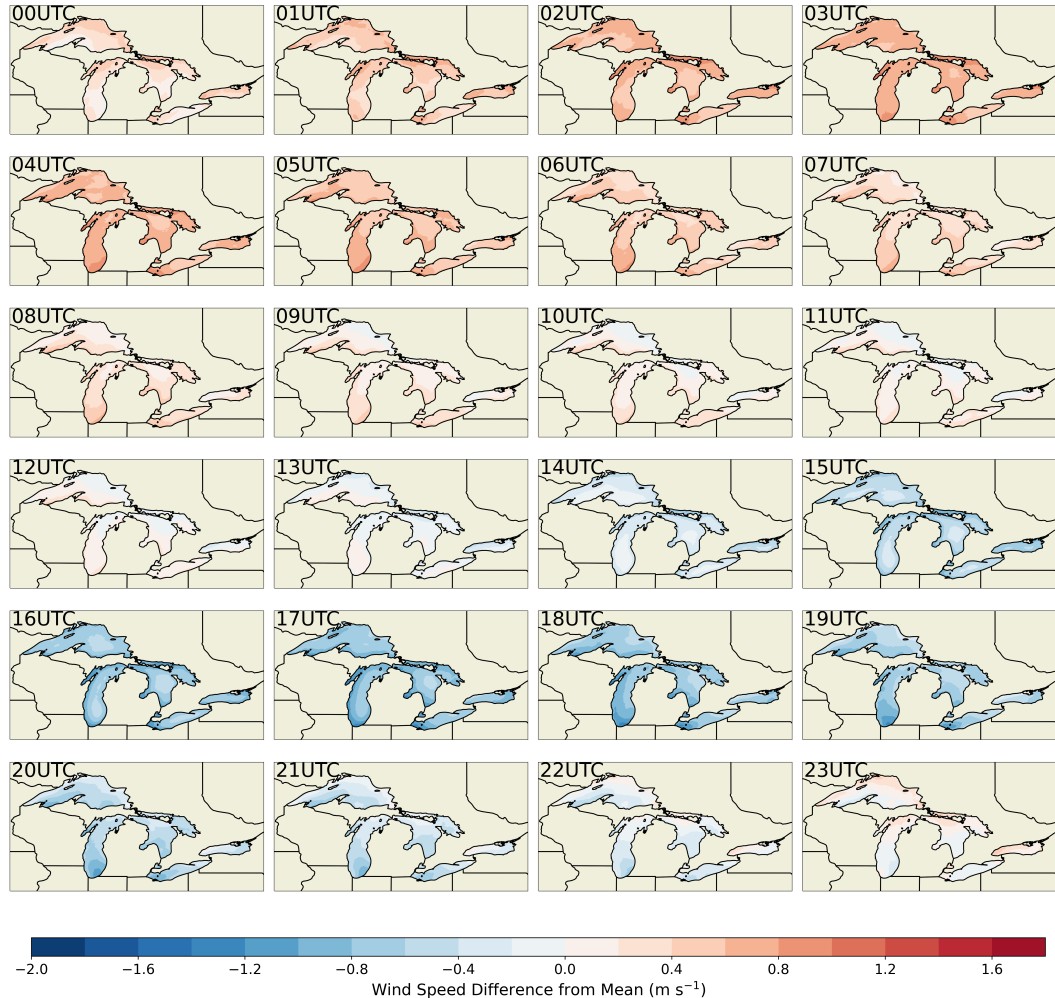

**Figure A10.** Maps showing the diurnal cycle in 160 m wind speed, expressed in terms of the difference for each UTC hour's mean wind speed from the overall mean, for the Great Lakes region.

Bodini, N., Castagneri, S., and Optis, M.: Long-term uncertainty quantification in WRF-modeled offshore wind resource off the US Atlantic
        coast, Wind Energy Science, 8, 607–620, https://doi.org/10.5194/wes-8-607-2023, 2023.

de Velasco, G. G. and Winant, C. D.: Seasonal patterns of wind stress and wind stress curl over the Gulf of Mexico, Journal of Geophysical
        Research: Oceans, 101, 18 127–18 140, https://doi.org/10.1029/96JC01442, 1996.

Donlon, C. J., Martin, M., Stark, J., Roberts-Jones, J., Fiedler, E., and Wimmer, W.: The operational sea surface temperature and sea ice
analysis (OSTIA) system, Remote Sensing of Environment, 116, 140–158, 2012.

Dörenkämper, M., Olsen, B. T., Witha, B., Hahmann, A. N., Davis, N. N., Barcons, J., Ezber, Y., García-Bustamante, E., González-Rouco,
        J. F., Navarro, J., et al.: The making of the New European Wind Atlas–part 2: Production and evaluation, Geoscientific model development,
        13, 5079–5102, https://doi.org/10.5194/gmd-2020-23, 2020.

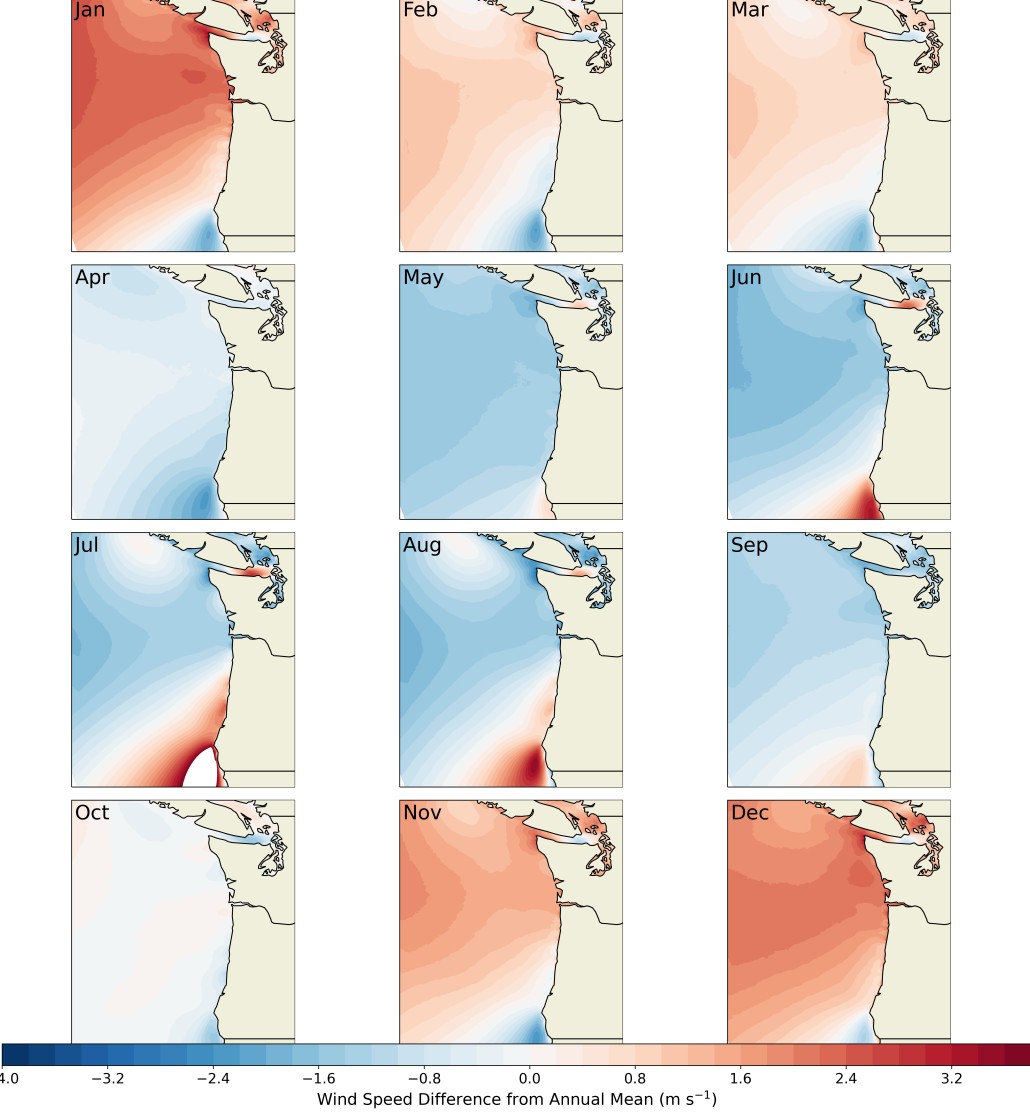

**Figure A11.** Maps showing the annual cycle in 160 m wind speed, expressed in terms of the difference for each month's mean wind speed from the overall mean, for the North Pacific region.

Draxl, C., Clifton, A., Hodge, B.-M., and McCaa, J.: The Wind Integration National Dataset (WIND) Toolkit, Applied Energy, 151, 355 –
366, https://doi.org/10.1016/j.apenergy.2015.03.121, 2015.

Fernando, H., Mann, J., Palma, J., Lundquist, J. K., Barthelmie, R. J., Belo-Pereira, M., Brown, W., Chow, F., Gerz, T., Hocut, C., et al.:
    The Perdigao: Peering into microscale details of mountain winds, Bulletin of the American Meteorological Society, 100, 799–819,
    https://doi.org/10.1175/BAMS-D-17-0227.1, 2019.



**Figure A12.** Maps showing the diurnal cycle in 160 m wind speed, expressed in terms of the difference for each UTC hour's mean wind speed from the overall mean, for the North Pacific region.



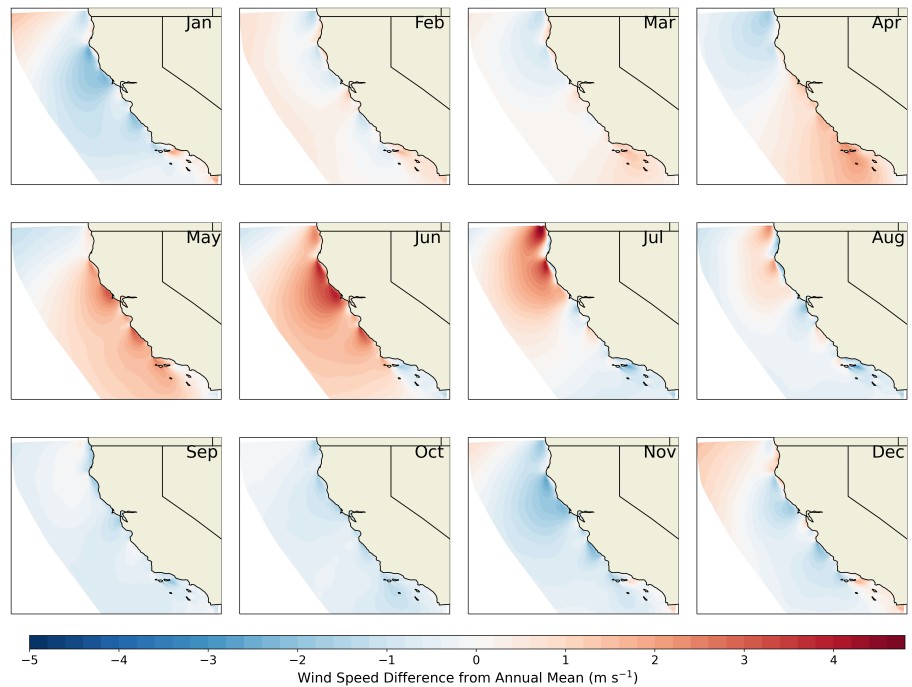

**Figure A13.** Maps showing the annual cycle in 160 m wind speed, expressed in terms of the difference for each month's mean wind speed from the overall mean, for the South Pacific region.

Fitch, A. C., Olson, J. B., Lundquist, J. K., Dudhia, J., Gupta, A. K., Michalakes, J., and Barstad, I.: Local and mesoscale impacts of wind
farms as parameterized in a mesoscale NWP model, Monthly Weather Review, 140, 3017–3038, https://doi.org/10.1175/MWR-D-11-00352.1, 2012.

Gelaro, R., McCarty, W., Suárez, M. J., Todling, R., Molod, A., Takacs, L., Randles, C. A., Darmenov, A., Bosilovich, M. G., Reichle, R.,
Wargan, K. amd Coy, L., Cullather, R., Draper, C., Akella, S., Buchard, V., Conaty, A., da Silva, A. M., Gu, W., Kim, G.-K., Koster, R.,
Lucchesi, R., Merkova, D., Nielsen, J. E., Partyka, G., Pawson, S., Putman, W., Rienecker, M., Schubert, S. D., Sienkiewicz, M., and
Zhao: The Modern-Era Retrospective Analysis for Research and Applications, version 2 (MERRA-2), Journal of Climate, 30, 5419–5454,
https://doi.org/https://doi.org/10.1175/JCLI-D-16-0758.1, 2017.

Grell, G. A., Dudhia, J., Stauffer, D. R., et al.: A description of the fifth-generation Penn State/NCAR Mesoscale Model (MM5), http:
//danida.vnu.edu.vn/cpis/files/Books/MM5%20Discription%20-%201995.pdf, 1994.

Hahmann, A. N., Sīle, T., Witha, B., Davis, N. N., Dörenkämper, M., Ezber, Y., García-Bustamante, E., González-Rouco, J. F., Navarro,
J., Olsen, B. T., et al.: The making of the New European Wind Atlas–part 1: Model sensitivity, Geoscientific model development, 13,
5053–5078, https://doi.org/10.5194/gmd-2019-349, 2020.

Hersbach, H., Bell, B., Berrisford, P., Hirahara, S., Horányi, A., Muñoz-Sabater, J., Nicolas, J., Peubey, C., Radu, R., Schepers, D., Sim-
mons, A., Soci, C., Abdalla, S., Abellan, X., Balsamo, G., Bechtold, P., Biavati, G., Bidlot, J., Bonavita, M., De Chiara, G., Dahlgren,
P., Dee, D., Diamantakis, M., Dragani, R., Flemming, J., Forbes, R., Fuentes, M., Geer, A., Haimberger, L., Healy, S., Hogan, R. J.,
Hólm, E., Janisková, M., Keeley, S., Laloyaux, P., Lopez, P., Lupu, C., Radnoti, G., de Rosnay, P., Rozum, I., Vamborg, F., Vil-



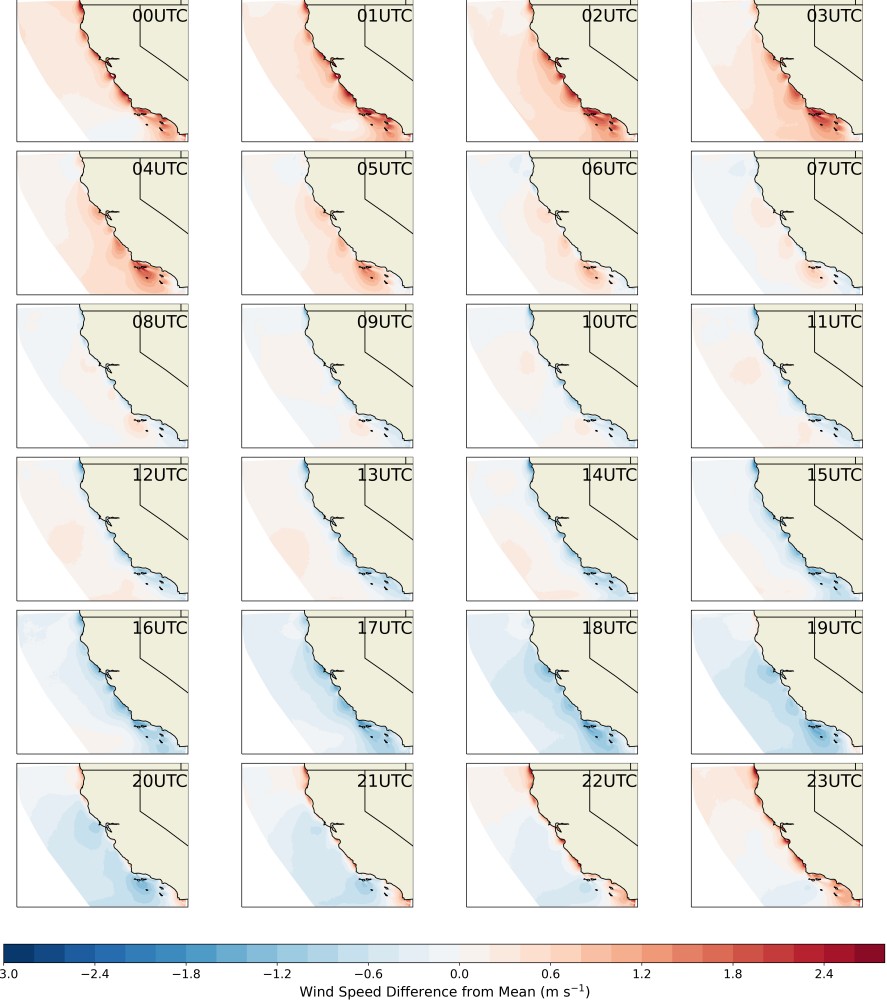

**Figure A14.** Maps showing the diurnal cycle in 160 m wind speed, expressed in terms of the difference for each UTC hour's mean wind speed from the overall mean, for the South Pacific region.

laume, S., and Thépaut, J.-N.: The ERA5 global reanalysis, Quarterly Journal of the Royal Meteorological Society, 146, 1999–2049, https://doi.org/https://doi.org/10.1002/qj.3803, 2020.

Hirahara, S., Balmaseda, M. A., Boisseson, E., and Hersbach, H.: 26 sea surface temperature and sea ice concentration for ERA5, Eur. Centre Medium Range Weather Forecasts, Berkshire, UK, ERA Rep. Ser, 26, https://www.ecmwf.int/sites/default/files/elibrary/2016/16555-sea-surface-temperature-and-sea-ice-concentration-era5.pdf, 2016.


Hong, S.-Y., Noh, Y., and Dudhia, J.: A new vertical diffusion package with an explicit treatment of entrainment processes, Monthly weather review, 134, 2318–2341, 2006.

Jiménez, P. A., Dudhia, J., González-Rouco, J. F., Navarro, J., Montávez, J. P., and García-Bustamante, E.: A revised scheme for the WRF surface layer formulation, Monthly weather review, 140, 898–918, https://doi.org/10.1175/MWR-D-11-00056.1, 2012.

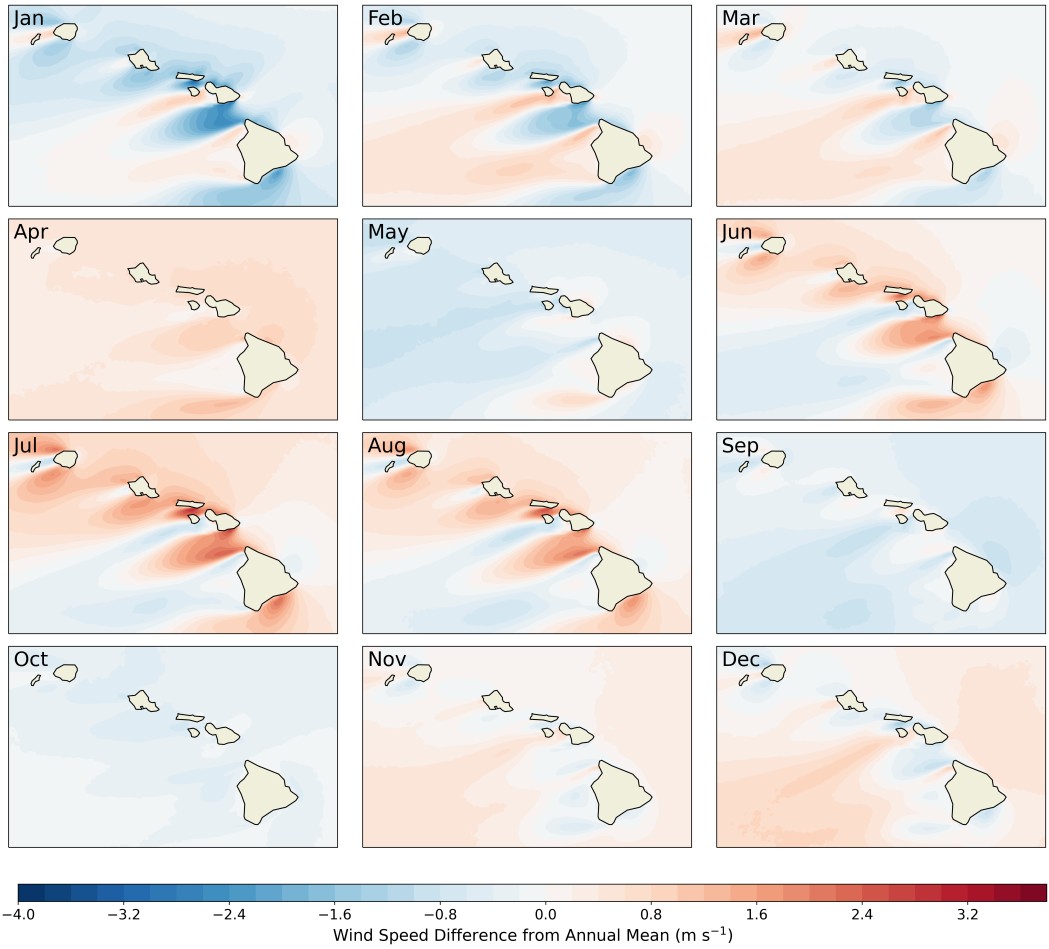

**Figure A15.** Maps showing the annual cycle in 160 m wind speed, expressed in terms of the difference for each month's mean wind speed from the overall mean, for the Hawaii region.

Juliano, T. W., Kosović, B., Jiménez, P. A., Eghdami, M., Haupt, S. E., and Martilli, A.: "Gray Zone" simulations using a three-dimensional planetary boundary layer parameterization in the Weather Research and Forecasting Model, Monthly Weather Review, 150, 1585–1619, https://doi.org/10.1175/MWR-D-21-0164.1, 2022.

Kosović, B., Munoz, P. J., Juliano, T., Martilli, A., Eghdami, M., Barros, A., and Haupt, S.: Three-dimensional planetary boundary layer parameterization for high-resolution mesoscale simulations, in: Journal of Physics: Conference Series, vol. 1452, p. 012080, IOP Publishing,
https://doi.org/10.1088/1742-6596/1452/1/012080, 2020.

Krishnamurthy, R., García Medina, G., Gaudet, B., Gustafson Jr., W. I., Kassianov, E. I., Liu, J., Newsom, R. K., Sheridan, L. M., and Mahon, A. M.: Year-long Buoy-Based Observations of the Air–Sea Transition Zone off the U.S. West Coast, Earth System Science Data Discussions, 2023, 1–53, https://doi.org/10.5194/essd-2023-115, 2023.

**Figure A16.** Maps showing the diurnal cycle in 160 m wind speed, expressed in terms of the difference for each UTC hour's mean wind speed from the overall mean, for the Hawaii region.

Nakanishi, M. and Niino, H.: Development of an improved turbulence closure model for the atmospheric boundary layer, Journal of the

Meteorological Society of Japan. Ser. II, 87, 895–912, https://doi.org/10.2151/jmsj.87.895, 2009.

Niu, G.-Y., Yang, Z.-L., Mitchell, K. E., Chen, F., Ek, M. B., Barlage, M., Kumar, A., Manning, K., Niyogi, D., Rosero, E., et al.: The community Noah land surface model with multiparameterization options (Noah-MP): 1. Model description and evaluation with local-scale measurements, Journal of Geophysical Research: Atmospheres, 116, 2011.

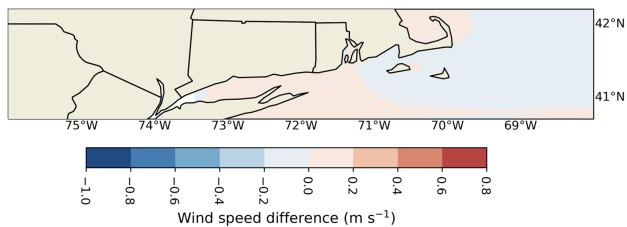

**Figure B1.** Map showing the difference in mean 160 m wind speed between the North Atlantic regional data set and the mid-Atlantic one, in the region where their WRF domains overlap.

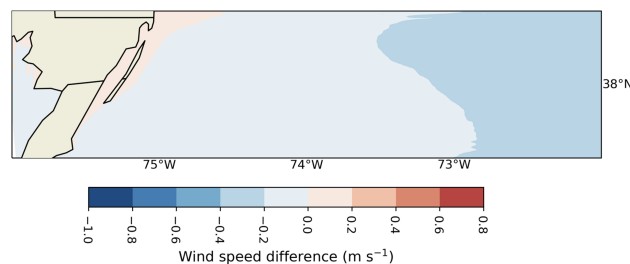

**Figure B2.** Map showing the difference in mean 160 m wind speed between the mid-Atlantic regional data set and the South Atlantic one, in the region where their WRF domains overlap.

Olson, J. B., Smirnova, T., Kenyon, J. S., Turner, D. D., Brown, J. M., Zheng, W., and Green, B. W.: A description of the MYNN surface-layer
scheme, https://doi.org/10.25923/f6a8-bc75, 2021.

Optis, M., Bodini, N., Debnath, M., and Doubrawa, P.: Best Practices for the Validation of U.S. Offshore Wind Resource Models, Tech. rep.,
National Renewable Energy Laboratory (NREL), Golden, CO (United States), https://doi.org/https://doi.org/10.2172/1755697, 2020a.

Optis, M., Kumler, A., Scott, G. N., Debnath, M. C., and Moriarty, P. J.: Validation of RU-WRF, the custom atmospheric mesoscale model of
the Rutgers Center for Ocean Observing Leadership, Tech. rep., National Renewable Energy Lab.(NREL), Golden, CO (United States),
https://doi.org/10.2172/1599576, 2020b.

Optis, M., Rybchuk, O., Bodini, N., Rossol, M., and Musial, W.: 2020 Offshore Wind Resource Assessment for the Cal-
ifornia Pacific Outer Continental Shelf, Tech. rep., National Renewable Energy Lab.(NREL), Golden, CO (United States),
https://doi.org/https://doi.org/10.2172/1677466, 2020c.

Optis, M., Rybchuk, O., Bodini, N., Rossol, M., and Musial, W.: Offshore Wind Resource Assessment for the California Pacific Outer Con-
tinental Shelf (2020), Tech. rep., National Renewable Energy Lab.(NREL), Golden, CO (United States), https://doi.org/10.2172/1677466,
2020d.

Pronk, V., Bodini, N., Optis, M., Lundquist, J. K., Moriarty, P., Draxl, C., Purkayastha, A., and Young, E.: Can reanalysis products outperform
mesoscale numerical weather prediction models in modeling the wind resource in simple terrain?, Wind Energy Science, 7, 487–504,
https://doi.org/10.5194/wes-7-487-2022, 2022.

Rosencrans, D., Lundquist, J. K., Optis, M., Rybchuk, A., Bodini, N., and Rossol, M.: Annual Variability of Wake Impacts on Mid-Atlantic
Offshore Wind Plant Deployments, Wind Energy Science Discussions, 2023, 1–39, https://doi.org/10.5194/wes-7-2085-2022, 2023.

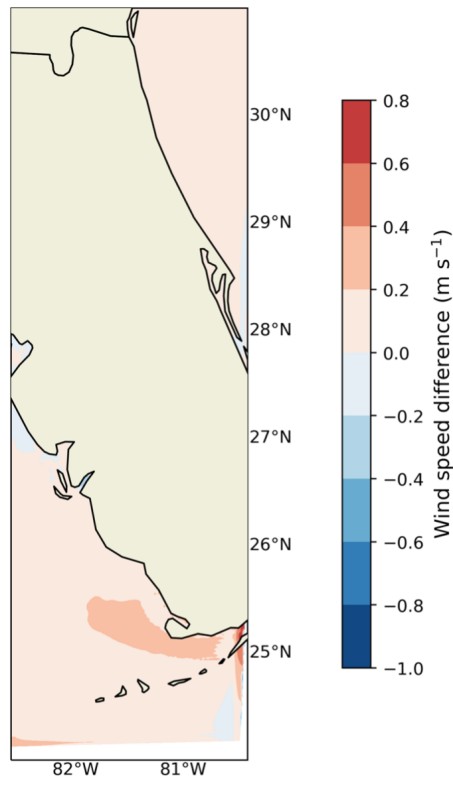

**Figure B3.** Map showing the difference in mean 160 m wind speed between the South Atlantic regional data set and the Gulf of Mexico one, in the region where their WRF domains overlap.

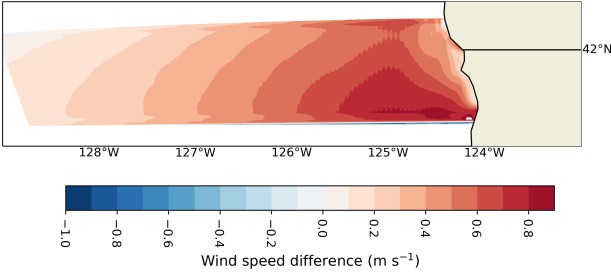

**Figure B4.** Map showing the difference in mean 160 m wind speed between the North Pacific regional data set and the South Pacific one, in the region where their domains overlap.

Rybchuk, A., Juliano, T. W., Lundquist, J. K., Rosencrans, D., Bodini, N., and Optis, M.: The sensitivity of the Fitch wind farm parameterization to a three-dimensional planetary boundary layer scheme, Wind Energy Science, 7, 2085–2098, https://doi.org/10.5194/wes-7-2085-2022, 2022.

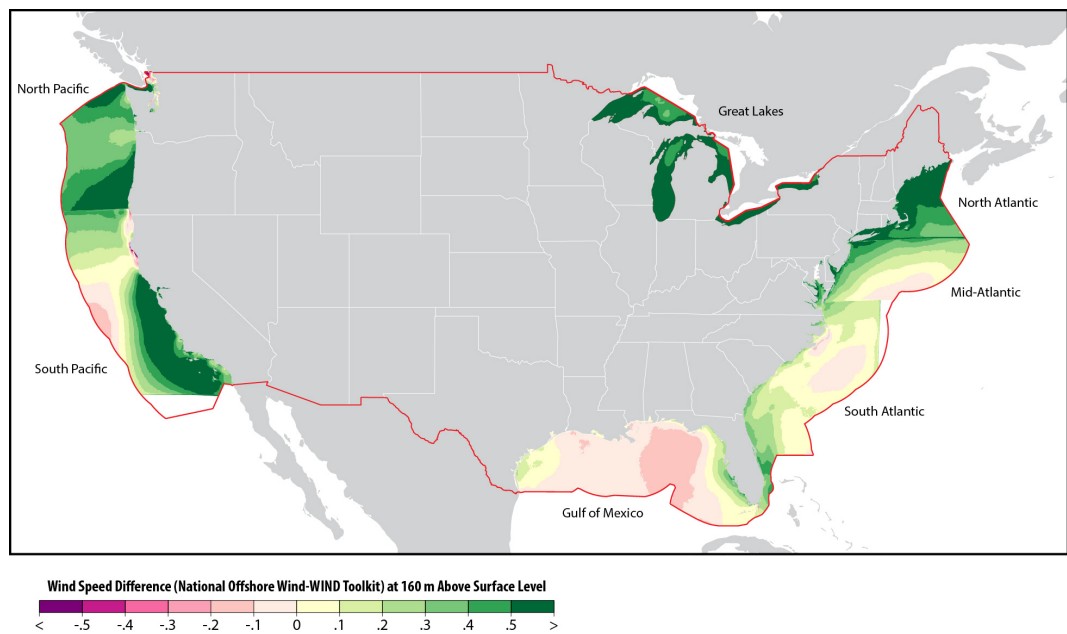

**Figure C1.** Map showing the difference in mean 160 m wind speed between the NOW-23 data set and the WIND Toolkit, for the EEZ in the contiguous U.S. where the two data sets overlap.

Shaw, W. J., Berg, L. K., Cline, J., Draxl, C., Djalalova, I., Grimit, E. P., Lundquist, J. K., Marquis, M., McCaa, J., Olson, J. B., et al.: The second wind forecast improvement project (WFIP2): general overview, Bulletin of the American Meteorological Society, 100, 1687–1699, https://doi.org/10.1175/BAMS-D-18-0036.1, 2019.

Shaw, W. J., Draher, J., Garcia Medina, G., Gorton, A. M., Krishnamurthy, R., Newsom, R. K., Pekour, M. S., Sheridan, L. M., and Yang, Z.: General analysis of data collected from DOE lidar buoy deployments off Virginia and New Jersey, Tech. rep., Pacific Northwest National

Lab.(PNNL), Richland, WA (United States), https://doi.org/10.2172/1632348, 2020.

Skamarock, W. C., Klemp, J. B., Dudhia, J., Gill, D. O., Liu, Z., Berner, J., Wang, W., Powers, J. G., Duda, M. G., Barker, D. M., and Huang, X.-Y.: A Description of the Advanced Research WRF Model Version 4, p. 162, 2019.

Taylor, K. E.: Summarizing multiple aspects of model performance in a single diagram, Journal of Geophysical Research: Atmospheres, 106, 7183–7192, https://doi.org/https://doi.org/10.1029/2000JD900719, 2001.

Thiébaux, J., Rogers, E., Wang, W., and Katz, B.: A new high-resolution blended real-time global sea surface temperature analysis, Bulletin of the American meteorological Society, 84, 645–656, https://doi.org/10.1175/BAMS-84-5-645, 2003.

Viselli, A., Filippelli, M., Pettigrew, N., Dagher, H., and Faessler, N.: Validation of the first LiDAR wind resource assessment buoy system offshore the Northeast United States, Wind Energy, 22, 1548–1562, https://doi.org/10.1002/we.2387, 2019.

Viselli, A., Faessler, N., and Filippelli, M.: LiDAR Measurements of Wind Shear Exponents and Turbulence Intensity Offshore the Northeast

United States, Journal of Offshore Mechanics and Arctic Engineering, 144, 042 001, https://doi.org/10.1115/1.4053583, 2022.





Wilczak, J., Finley, C., Freedman, J., Cline, J., Bianco, L., Olson, J., Djalalova, I., Sheridan, L., Ahlstrom, M., Manobianco, J., et al.: The Wind Forecast Improvement Project (WFIP): A public–private partnership addressing wind energy forecast needs, Bulletin of the American Meteorological Society, 96, 1699–1718, https://doi.org/10.1175/BAMS-D-14-00107.1, 2015.

Zavala-Hidalgo, J., Romero-Centeno, R., Mateos-Jasso, A., Morey, S. L., and Martínez-López, B.: The response of the Gulf of Mex-
ico to wind and heat flux forcing: What has been learned in recent years?, Atmósfera, 27, 317–334, https://doi.org/10.1016/S0187-6236(14)71119-1, 2014.