# Peer review of "The 2023 National Offshore Wind data set (NOW-23)"

_Earth System Science Data, 2023_

## Author Comment (AC1)

*In this document, the reviewer's comments are in black, the authors' responses are in red.*

We thank the reviewer for their thoughtful comments, which gave us an opportunity to revisit our analysis.

**General comments**

This article details a recently developed offshore wind data set, the NOW-23 data set, aimed at wind climatology studies for power generation, covering most offshore regions around the contiguous USA and the Great Lakes. The manuscript outlines the data set, the processes used to derive the data from numerical modelling, and details the choices (and validation of these choices) in model set up and execution. The authors conclude with a summary of uncertainties in the data set and include a brief description of an accompanying data set, NOW-WAKES.

The manuscript is well written and concise, however, perhaps in their attempt at brevity the authors have missed the opportunity to reinforce their reasoning for producing this work (see comment #1 below). I do not believe any extra sections are required, but some statistics regarding the improved performance of NOW-23 when compared to the previous version (WIND) would help cement the justification for the manuscript. This is partly touched upon in Appendix C, but a table would help clarify these differences.

Regarding the data set, I was able to access the online repository and download a sample set of data (~1.5 Gb), extract it (in the form of a CSV file), and view/analyze the data.

I recommend publication after the authors consider making some minor changes in response to the comments below.

**Specific comments - Manuscript**

1. Section 1, lines 35 - 40: Not all readers may be aware of the shortcomings of WIND, therefore stating explicitly how long WIND was reforested for compared to the new 20-year period for NOW-23 would be advantageous. Either this could be added to the text, or perhaps the authors could add a table (or modify Table 1.) for the reader to easily to compare these parameters between WIND and NOW-23.
   A new table could detail the differences between WIND and NOW-23 in:
   - Temporal Resolution
   - Forecast period
   - Horizontal resolution
   - Heights
   - Etc

   This is a great point, we have modified Table 1 to include WIND Toolkit attributes, and have expended the list of attributes listed to make sure we can show all the differences between the two data sets in one place:

**Table 1.** Main attributes of the NOW-23 data set, compared to those used in the older WIND Toolkit. Attributes in **bold** are the result of a region-specific sensitivity analysis in NOW-23.

| | **NOW-23** | **WIND Toolkit** |
|---|---|---|
| **Temporal extent** | 2000–2019/22 (varies with region) | 2007–2013 |
| **WRF model version** | 4.2.1 | 3.4.1 |
| **Nesting** | 6 km, 2 km | 18 km, 6 km, 2 km |
| **Temporal resolution (output)** | 5-min | 5-min |
| **Vertical levels** | 61 | 41 |
| **Near-surface-level heights** | 12 m, 34 m, 52 m, 69 m, 86 m, 107 m, 134 m, 165 m, 200 m | 15 m, 47 m, 80 m, 112 m, 145 m, 177 m |
| **Reanalysis forcing** | **ERA-5** | ERA-Interim |
| **Sea surface temperature forcing** | **OSTIA** | ERA-Interim |
| **Atmospheric nudging** | Spectral nudging on 6-km domain, applied every 6 hours[1] | Spectral nudging on 18-km and 6-km domains, applied every 6 hours |
| **Planetary boundary layer scheme** | **MYNN or YSU** (varies with region) | YSU |
| **Surface layer scheme** | **MYNN or MM5** (varies with region) | MM5 |
| **Land surface model** | **NOAH** | NOAH |
| **Microphysics** | Ferrier | |
| **Longwave radiation** | Rapid Radiative Transfer Model | |
| **Shortwave radiation** | Rapid Radiative Transfer Model | |
| **Topographic database** | Global Multi-Resolution Terrain Elevation Data from the United States Geological Service | |
| **Land-use data** | Moderate Resolution Imaging Spectroradiometer 30s | |
| **Cumulus parameterization** | Kain-Fritsch | |

2. I found the approach of validating each area independently using the WRF ensembles as thorough and exhaustive, therefore improving the end user's confidence in the data set. Additionally, section 11 describes well the uncertainty in the modelled wind speed. However, this also raises a question: how 'bad' would the ensemble members have to be compared to observations for you not produce a product for a certain region? Has this been considered? I feel it not addressed in the manuscript.

   This is an interesting point, and we think it really depends on the specific application each user has in mind for the data set. We were funded to create a full data set for all U.S. offshore waters. Ideally, one would have uncertainty data for all modeled regions, so that each user can make their informed decisions on whether the level of uncertainty in a given region is acceptable for their specific application. Unfortunately, our funding (combined with the limited availability of observations) only allowed us to explore the uncertainty piece in the Mid-Atlantic region, which is however the most important region for upcoming offshore wind energy development. Hopefully, future funding will be available to complete the uncertainty analysis for the remaining regions, too. We have added a couple of sentences at the beginning of section 11 to reflect this idea.

3. Is Section 12 necessary if two previously published articles already detail this information? If word-count is at a premium then this could be a section to consolidate, using perhaps just the first paragraph to outline the product and draw the reader's attention to the other two articles.

   We have included this section to maximize the exposure of all the components of the NOW-23 data set (also considering that word count is not a constraint for ESSD). Still, we have now shortened this section, to keep the focus of the paper on the main, long-term component of NOW-23.

4. I applaud the use the Taylor diagrams. The clear and concise explaination of what would be a perfect score is greatly appreciated by those of us who do not use these diagrams regularly.

Thank you!

5.  Section 13, line 575: is it possible the term 'long-term data' should be changed to 'Sample data' to match the nomenclature in the online data repository?
    Done!

6.  I'm not sure the Appendix figures were meant to be interspersed throughout the References. I am sure the typesetter will fix this, but it is worth double checking to avoid it happening again.
    We agree the current layout is less than ideal. We used Copernicus' template (and did not want to change it), so we will follow the instructions from the typesetter.

**Specific comments – Data set**

7.  On the NOW-23 repository page, I would not say the statement *'Examples of using the HSDS Service to Access NOW-23 data. Contains resources that will help users view the data found in this submission.'* is accurate. The link that opened for me was for the WIND Toolkit page in the rex documentation. It may be true that these resources will also allow users to explore the NOW-23 dataset, however, since this is a substantial update of the resource, I would expect the link to send me to a dedicated page detailing how to use it specifically for the NOW-23 dataset. Even if the page is simply copied and elements of the text are changed to make it relevant to the NOW-23 dataset, it may not be necessary to construct an entirely new page. It would be pertinent to update other pages in the rex documentation to reflect its use with NOW-23 (e.g., the home page).
    Great point, and great timing. NREL's rex team is in the process of updating and improving the documentation page. This update will be completed in the next few weeks, and the updated documentation page will more generally refer to all the NREL-produced resource data sets (including NOW-23), which all share the same data structure.

8.  What has been done about outliers in the dataset? The first sample data I opened ('Virginia_lidar') had a boundary layer height maximum of 6553.5 m, which appeared, when compared to the rest of the sample data, as definitely an outlier (but I could be wrong).
    We have decided to keep all WRF output data in the published files for several reasons. First, defining outliers is a tricky aspect, so that different users might have different preferred definitions. Also, modeled data sets are expected to be continuous in space and time, so that introducing gaps in the data could complicate potential applications from the user base. Still, we agree this should be pointed out, and so we have added the following statement to the OpenEI page: "No filters have been applied to the raw WRF output.".

**Technical corrections**

9.  Table 14: Should this entry for WRF4 match the WRF4 from either Tables 12 or 13? If so, LSM should be NOAH-MP. At present, it appears to match the entry in Table 12 for WRF2.
    Thanks for catching this typo! We have updated this in the revised version of the tables (now Tables 2 and 3).

10. Figure 18: Please add the period covered in the caption (I.e., is it for the entire 20 yr. period?).
    We have added the following: ", calculated using the full temporal extent of each data set".

11. Figure B4: add 'WRF" to match the other figure captions of the same nature.
    Done!

12. Figure C1: same comment as #10. Is this for the entire period for each data set, or a shorter period where there is concurrent data.
    We have added the following: "The difference is calculated using the full temporal extent of each data set."

**Reference corrections**
13. Shamarock et al. (2019): is there a DOI for this entry available?
    We have added the DOI.
14. Wilczark et al. (2015): I believe all authors must be listed (although I assume the copy editor will confirm this).
    We have added all authors to the .bib file, and will let the Copernicus copy editor confirm the preferred approach according to the ESSD template.
15. Hahmann et al. (2020): same comment as above
    Same as above.
16. Krishnamurthy et al. (2023): is there an updated entry for this reference?
    Yes, the paper is now published, and we updated the reference accordingly.

---

## Author Comment (AC2)

*In this document, the reviewer's comments are in black, the authors' responses are in red.*

We thank the reviewer for their thoughtful comments, which gave us an opportunity to revisit our analysis.

**Scope**
The scope of the manuscript is well-suited for this journal.

**Originality**
The question of originality is not critical for a data journal. However, this is very important, and it would encourage other groups to publish the description of their datasets in this manner.

**Scientific rigor**
In general, the scientific rigor is adequate. However, I have a few questions and clarification points below.

1. The wind time series uses a 5-minute temporal resolution. Have you checked that the spectra contain energy at this time scale? If not, this should be mentioned somewhere.
   The choice of the 5-minute temporal resolution comes from the needs of the grid integration community. We have now added a sentence to mention this ("The choice of the 5-minute temporal resolution is also dictated to accommodate needs of the grid integration community."). Regarding spectra, we have not checked those for NOW-23, but we have relied on the analysis that was completed for the previous-generation WIND Toolkit (see Fig. 9 in http://dx.doi.org/10.1016/j.apenergy.2015.03.121), which did not show a significant peak at 5 minutes.

2. L104-109: The ERA5 reanalysis uses the OSTIA SST product. This should be mentioned, and unsurprisingly, forcing the WRF model with the same SST is often advantageous.
   We mention this aspect in Section 2: "The first SST product we consider is the Operational Sea Surface Temperature and Sea Ice Analysis (OSTIA) data set produced by the UK Met Office, which provides data at 1/20 deg horizontal resolution and is the standard product included in both ERA5 and MERRA-2."

3. How can you explain the observed wind profile in Figure 13? By the way, what is the source of SST for the Great Lakes? Are these points treated as lakes or seas?
   Unfortunately, the relatively limited (in time) and old nature of the Great Lakes data set limits our ability to be fully confident about the observed data: the shape of the mean wind profile could be due to poor QC of the raw lidar data (not performed by us and not accessible to us) or it could be a physical mechanism (e.g., a low-level jet observed on average at that location over the period of record). Regarding the SST data for the region, the same data set (OSTIA) is used. We have clarified this in the text.

4. L380. It is not clear what "overestimates atmospheric stability" means. More stable? Values of temperature gradients? Please clarify.
   We have changed it "overestimates the frequency of stable conditions".

**Writing**
The writing is clear and well-structured.

**Length**

The manuscript is a bit long.  Also, the structure is tedious, looking more like a report than a scientific paper. I suggest removing the sub-sections for each zone, which often contain only one or two sentences. The eight regions could be combined according to each ocean basin.

We have removed all sub-sections as suggested. We have also merged most tables into two (see answer to one of the comments below). As a result, the article is now 6 pages shorter. We still kept the current divisions into 8 regions to follow the division of the NOW-23 data set into 8 modeling domains.

**Figures and tables**

I suggest redrawing most of the wind profiles and Taylor diagrams.  It is not possible to distinguish the various simulations.  I suggest using a logarithmic y scale in the profiles, which is standard in wind energy. For the Taylor diagrams, it is possible to zoom in the relevant region so that the differences between the runs are highlighted. See, for example, https://agupubs.onlinelibrary.wiley.com/doi/10.1002/2017JD027504.

We have updated the figures as suggested.

Are tables 2 and 12 the same? Could the tables be compressed to shorten and facilitate reading? I suggest one table with all the possible ensemble members and a final column stating which domain they are used in.

We have merged all model setup tables into two tables (now Tables 2 and 3).

I also suggest that all figure captions contain the height and time period used in the validation (e.g., Figs 3, 4, 8, 9, 11). The period needs to be added in the caption of Figure 18.

We have updated the figure captions.

**Title**

The title is short and informative.

**Abstract**

The abstract is concise but contains most of the relevant information. The years and the model used in the simulations should also be included to facilitate future searches.

We have added the year and model information to the abstract.

**References**

The references are relevant and comprehensive.

**Recommendation**

I recommend publishing the article once the suggestions regarding the structure and the figures' redrawing are considered.

**Minor corrections:**

1. Please substitute "WRF" with "WRF model" or "the WRF model" when appropriate.
   Changed.
2. L79: consider -> considered. Maybe the rest of the sentence should also be in the past?
   Changed.